# Direct visualization of HIV-1 core nuclear import and its interplay with the nuclear pore

Zhen Hou [1,6], Stanley Fronik [1,2,6], Yao Shen [1,6], Long Chen[1], Christopher Thompson [3], Sarah Neumann[3] & Peijun Zhang [1,4,5 ✉]

## Abstract

**Direct visualization of HIV-1 nuclear import through the nuclear pore complex (NPC) presents a technical challenge due to the rarity of this process. To enable systematic investigation, we developed a robust in situ system that mimics HIV-1 nuclear import in a near-native context using isolated HIV-1 virus like particles (VLP) cores and permeabilized CD4 + T lymphocyte (CEM) cells. This approach supports docking and translocation of abundant viral cores through nuclear pores into the nucleus. For high-resolution visualization, we implemented an integrated correlative approach to guide cryo-focused ion beam (cryo-FIB) milling and cryo-electron tomography (cryo-ET) imaging, enabling precise targeting and structural characterization of individual nuclear import events. Using this workflow, we visualized 510 HIV-1 VLP cores at distinct stages of nuclear import, capturing key snapshots of the full progression of nuclear import. Subsequent statistical and structural analyses allow classification of core morphologies and identification of translocation-associated remodeling in nuclear pores. This work provides a methodological foundation for dissecting HIV-1 and potentially other viruses nuclear import processes and post-entry events in a controlled and quantitative manner.**

**Keywords** HIV-1 Nuclear Import; Nuclear Pore; Cryo-CLEM; Cryo-ET and Subtomogram Averaging; Cryo-FIB/SEM
**Subject Categories** Microbiology, Virology & Host Pathogen Interaction; Structural Biology

## Introduction

Human immunodeficiency virus type 1 (HIV-1), a member of the lentivirus family can infect non-dividing cells, including resting CD4[+] T cells and macrophages, which are key components of the immune system (Dai et al, 2009; Gartner et al, 1986; Guedan et al,

2021). This distinguishes it from other retroviruses like murine leukemia virus (MLV), which rely on nuclear envelope breakdown during mitosis to access the host genome (Matreyek and Engelman, 2013). In contrast, HIV-1 can traverse an intact nuclear envelope via the nuclear pore complex (NPC) and integrate its genetic material into host chromatin, enabling persistent infection (Guedan et al, 2021).

After fusion of the viral envelope with the host cell membrane, the HIV-1 core is released into the cytoplasm (Shen et al, 2021). This core consists of approximately 250 hexamers and 12 pentamers of capsid (CA) proteins (Ganser et al, 1999; Welker et al, 2000), forming a fluorene cone structure (Byeon et al, 2009; Zhao et al, 2013) that encloses the viral RNA genome along with essential enzymes such as reverse transcriptase (RT) and integrase (IN). The capsid plays multiple critical roles throughout the HIV-1 life cycle: shielding the viral genome from host immune surveillance, facilitating reverse transcription, and mediating cytoplasmic trafficking and nuclear import (Alvarez et al, 2017; Liu et al, 2016; Zila et al, 2021b). Recent studies suggest that the HIV-1 capsid remains largely intact during nuclear import (Kreysing et al, 2025; Zila et al, 2021a); however, its molecular interplay with the NPC during nuclear import, and its post-entry uncoating, remains poorly understood.

The NPC is a complex structure composed of over 30 nucleoporins embedded within the nuclear envelope and regulates the bidirectional transport between the cytoplasm and the nucleus (Hampoelz et al, 2019; Wente and Rout, 2010). Within its center channel lies the FG-mesh, a selective permeability barrier formed by intrinsically disordered domains of nucleoporins enriched in phenylalanine-glycine (FG) repeats. While small molecules (<40 kDa) can diffuse passively through this mesh, the transport of larger complexes, such as ribonucleoproteins, requires active transport mediated by nuclear transport receptors (Ay et al, 2025; Gorlich and Kutay, 1999; Timney et al, 2016). The HIV-1 capsid interacts directly with specific nucleoporins, including Nup153 and Nup358, and engages with the FG-mesh to facilitate passage through the NPC (Bichel et al, 2013; Di Nunzio et al, 2012; Fu et al, 2024; Lin et al, 2013; Matreyek and Engelman, 2011; Matreyek et al, 2013; Schaller et al, 2011). Given that the HIV-1 capsid measures ~120 nm in length and ~60 nm in width (Welker et al, 2000), and

[1]Division of Structural Biology, Wellcome Trust Centre for Human Genetics, University of Oxford, Oxford, UK. [2]Section Electron Microscopy, Department of Cell and Chemical Biology, Leiden University Medical Center, Leiden, The Netherlands. [3]Materials & Structural Analysis, Thermo Fisher Scientific, Eindhoven, The Netherlands. [4]Diamond Light Source, Harwell Science and Innovation Campus, Didcot, UK. [5]Chinese Academy of Medical Sciences Oxford Institute, University of Oxford, Oxford, UK. [6]These authors contributed equally: Zhen Hou, Stanley Fronik, Yao Shen. ✉E-mail: peijun.zhang@strubi.ox.ac.uk

that the width is comparable to the NPC diameter in situ (Bley et al, 2022; Mosalaganti et al, 2022; Schuller et al, 2021; Singh et al, 2024; Zimmerli et al, 2021), it remains unclear how the viral core passes through such a narrow channel. Whether the capsid or the NPC undergoes structural remodeling during HIV-1 nuclear entry is still an open question.

Investigating the mechanisms of HIV-1 nuclear import in HIV-1-infected cells is challenging due to the low frequency of productive import events—only ~1 in 50 cytoplasmic viral complexes successfully reach the nucleus (Burdick et al, 2013). As a result, previous cryo-ET studies have been limited to analyzing only a small number of HIV-1 cores during nuclear import (Kreysing et al, 2025; Zila et al, 2021a). Moreover, in vitro reconstitution of NPCs is nearly impossible due to its structural complexity (Hampoelz et al, 2019) and NPCs from isolated nuclear envelopes are too narrow for HIV-1 cores (von Appen et al, 2015). To overcome these limitations, we established a robust in situ system that reconstitutes HIV-1 nuclear import by incubating purified HIV-1 VLP cores with permeabilized CD4$^+$ T cells. This system preserves native nuclear envelope integrity and significantly increases the frequency of import events. To enable targeted, high-resolution imaging of the HIV-1 core translocation through nuclear pores, we combined cryo-correlative light and electron microscopy (cryo-CLEM), cryo-focused ion beam (cryo-FIB) milling, and cryo-electron tomography (cryo-ET), which together allowed us to capture hundreds of HIV-1 VLP cores engaging with NPCs throughout the course of nuclear entry. Statistical and structural analyses revealed specific capsid morphologies preferentially undergoing import and remodeling events in the NPC during HIV-1 core translocation. Together, our work established a robust in situ system and a highly efficient correlative cryo-ET workflow, providing a powerful platform to uncover mechanistic insights into HIV-1 nuclear import and subsequent post-import events, and offering a broadly applicable approach for studying nuclear import processes more generally.

# Results and discussion

## Recapitulation of HIV-1 nuclear import using permeabilized T cells and isolated HIV-1 cores

The transient and infrequent nature of HIV-1 nuclear import events in infected cells poses a significant technical challenge for structural analysis (Burdick et al, 2017; Burdick et al, 2020; Francis and Melikyan, 2018; Zila et al, 2021a). To overcome this limitation, we established an in situ reconstituted system for HIV-1 nuclear import by combining CD4$^+$ T cell nuclei with purified HIV-1 VLP cores. To prepare structurally intact nuclei, we optimized two methods of nuclear isolation from CEM cells, a CD4$^+$ T cell line: mechanical lysis and digitonin permeabilization. Sample integrity was initially assessed using phase-contrast and fluorescence microscopy (Fig. EV1A), and the preservation of nuclear envelope and NPC barrier function was validated via exclusion of 150 kDa FITC-dextran (Raices and D'Angelo, 2022) under isotonic conditions (Fig. EV1B), confirming that the permeability barrier remained intact.

To obtain HIV-1 cores, we generated mNeonGreen-labeled integrase (mNeonGreen-IN) virus-like particles (VLPs) under biosafety-compliant conditions using the lentiviral packaging

vector PsPAX2 with a pVpr-mNeonGreen-IN plasmid. These VLPs mimic mature HIV-1 particles but are replication-deficient, as they lack both the viral genome and envelope proteins. A detergent-based "spin-thru" protocol followed by sucrose gradient ultracentrifugation (Aiken, 2009), yielded highly enriched cores, visible as a distinct fluorescent band (Fig. EV1C, inset). Cryo-EM imaging of VLP-derived cores revealed a ~ 1:1 ratio of cone- and tube-shaped morphologies, in contrast to the ~5:1 ratio typically seen in native virions (Fig. EV1D). Nonetheless, the sizes of cone-shaped and tube-shaped cores from VLPs are similar to the cone-shaped and tube-shaped cores from virions (Fig. EV1E).

When incubated with permeabilized CEM cells, mNeonGreen-IN-labeled cores localized robustly to the nuclear envelope regardless of lysis method (Figs. 1A and EV1F). Colocalization with the nucleoporin Nup358 revealed engagement with NPCs, with some cores migrating at distinct depths of NPCs, judging by the relative position between mNeonGreen-IN and anti-Nup358 signals (Fig. 1B). Importantly, cores were also observed within the nucleoplasm, confirming functional nuclear import (Figs. 1C and EV1F). This reconstituted system yielded high numbers of nucleus-associated cores, averaging 150 and 217 per nucleus following mechanical lysis and digitonin treatment, respectively (Fig. 1D). Compared to HIV-1-infected cells, where only ~1 in 50 viral particles successfully enter the nucleus (Burdick et al, 2013), our reconstituted system reproducibly allows for analysis of at least 100–200 cores interacting with each nucleus (Figs. 1D and EV1G). This substantial enrichment enables a high-throughput structural investigation of HIV-1 nuclear import. The observed accumulation of fluorescently labeled integrase (IN) at the nuclear envelope compared to the nucleoplasm further highlights the high affinity of HIV-1 core to NPCs and supports the notion that nuclear import represents a rate-limiting step in infection (Burdick et al, 2017; Burdick et al, 2020; Francis and Melikyan, 2018).

## High-throughput correlative cryo-ET workflow for capturing nuclear import

Direct imaging of HIV-1 nuclear import within the large volume (~10 μm) of a T cell nucleus requires a targeted strategy to ensure both efficient lamella preparation and accurate localization of HIV-1 cores (50–100 nm) within those lamellae. To this end, we developed a correlative imaging approach combining cryo-fluorescence microscopy (cryo-FM), targeted cryo-focused ion beam (cryo-FIB) milling, and cryo-ET guided by HIV-1 integrase (IN) fluorescence signals on lamellae (Fig. 2A,B). We used two complementary strategies to generate targeted cryo-FIB lamellae: one uses a planar lift-out method which allows wider (20–30 μm) and longer (15–20 μm) lamellae to be produced from flatter sample areas of interest (Fig. 2A) (Klumpe et al, 2022; Kuba et al, 2021; Mahamid et al, 2015; Parmenter and Nizamudeen, 2021; Rubino et al, 2012; Schaffer et al, 2019; Schiotz et al, 2024; Schreiber et al, 2018), while the other employs automated cryo-FIB to produce abundant correlated thin (100–200 nm) narrow (5–10 μm) lamellae for high throughput (Fig. 2B) (Arnold et al, 2016; Berger et al, 2023; Fu et al, 2019; Jun et al, 2011; Jun et al, 2013; Tacke et al, 2021; Wagner et al, 2020; Wang et al, 2012). Both mNeonGreen-IN signal which labels HIV-1 cores and SiR-DNA signal which marks the cell nucleus were used to guide the cryo-FIB milling positioning to specifically target HIV-1 cores associated with the nucleus

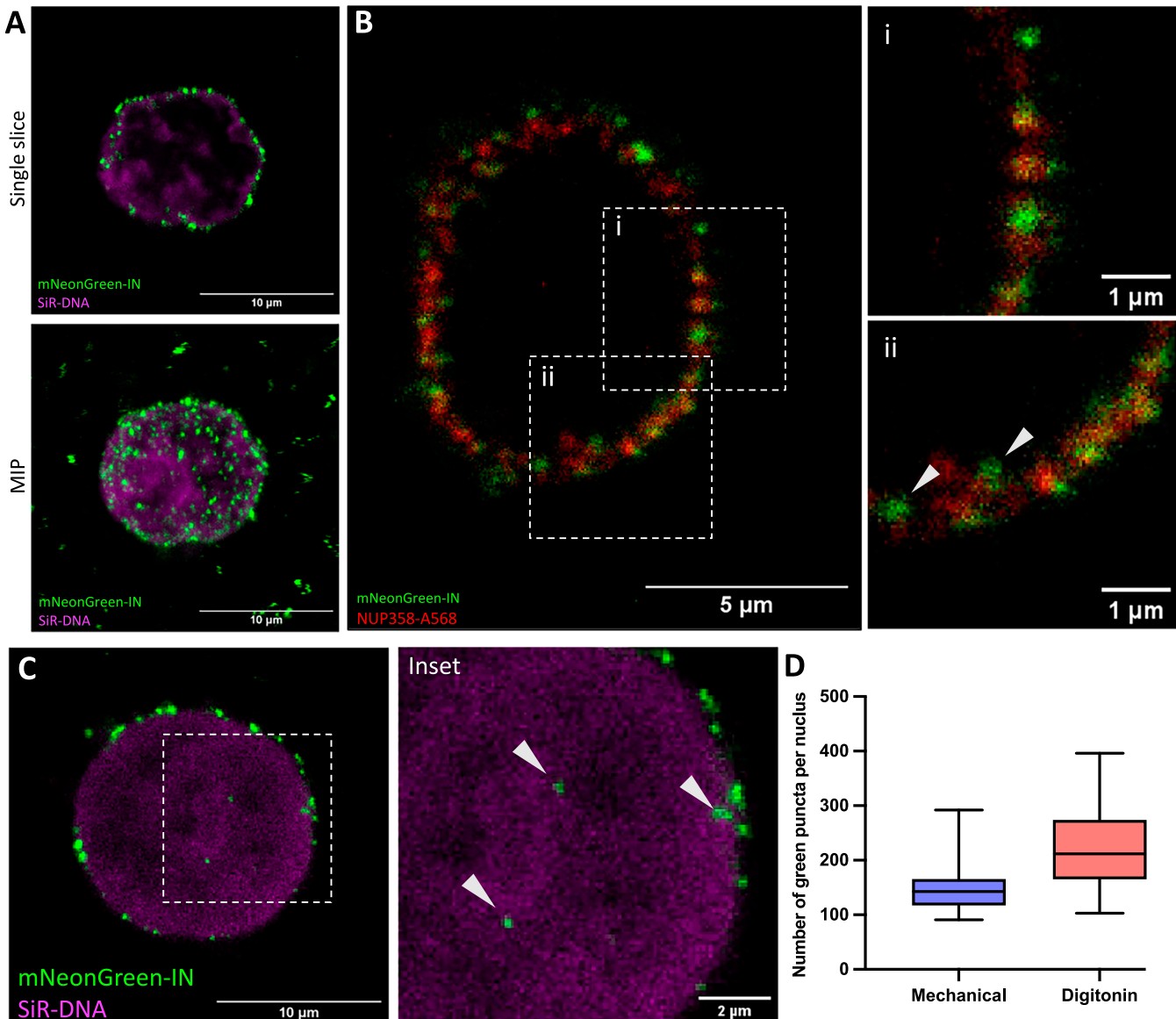

**Figure 1. Recapitulation of functional nuclear import of HIV-1.**

(**A**) Representative confocal images of mechanically permeabilized CEM cells mixed and incubated with isolated HIV-1 VLP cores. (green) mNeonGreen-IN labelled HIV-1 cores. (magenta) SiR-DNA labelled nucleus. Top: single Z-slice. Bottom: maximum intensity projection (MIP) of the combined Z-slices. (**B**) Representative confocal images of mechanically isolated CEM nuclei mixed and incubated with isolated HIV-1 cores followed by fixation and immunostaining of Nup358. (green) mNeonGreen-IN labelled HIV-1 cores. (red) αNup358-A568 labelled NPCs. Locations of insets are indicated by the white boxed areas. White arrowheads indicate mNeonGreen-IN signal inside the nucleus. (**C**) Representative confocal images of HIV-1 cores decorating CEM nuclei. Colours are the same as in (**A**). Inset: zoomed-in view of the white boxed area. Arrows indicate mNeonGreen-IN signal inside the nucleus. (**D**) A box plot representing the number of mNeonGreen-IN puncta decorating a single nucleus, either by mechanical lysis or by digitonin permeabilization of CEM cells. Nuclei from CEM cells obtained via mechanical lysis show 150 ± 48 puncta (left, *n* = 16), and nuclei obtained through digitonin permeabilization show 217 ± 77 puncta (right, *n* = 16). The centre line is the median, whiskers are min and max, and box bounds are the 25th and 75th percentiles. Data are pooled from two biological replicates (Fig. EV1G). Source data are available online for this figure.

(Fig. 2A,B). The final lamellae with a thickness ranging from 100 to 200 nm that contained detectable mNeonGreen-IN fluorescence signals were used for cryo-ET data collection (Fig. 2A,B). An illustrative tomogram collected at a lamella position where mNeonGreen-IN signal correlated, revealed clear ultrastructural features, including nuclear envelopes, NPCs, abundant ribosomes and nucleosomes (Fig. 2C,D). Interestingly, two successive HIV-1 cores were found at an NPC: one almost imported tube-shaped core

exiting the NPC, immediately followed by a cone-shaped core docking on the same NPC (Fig. 2C,D; Movie EV1).

This multi-modal cryo-CLEM workflow achieved a targeting efficiency over 50% (Fig. 3C), a substantial improvement over the previous 1–3% success rate observed in previous studies that lacked fluorescence guidance (Kreysing et al, 2025; Zila et al, 2021a). Among those 510 captured HIV-1 cores, over 86% were associated with the T cell nucleus (Fig. 3D), corroborating light microscopy

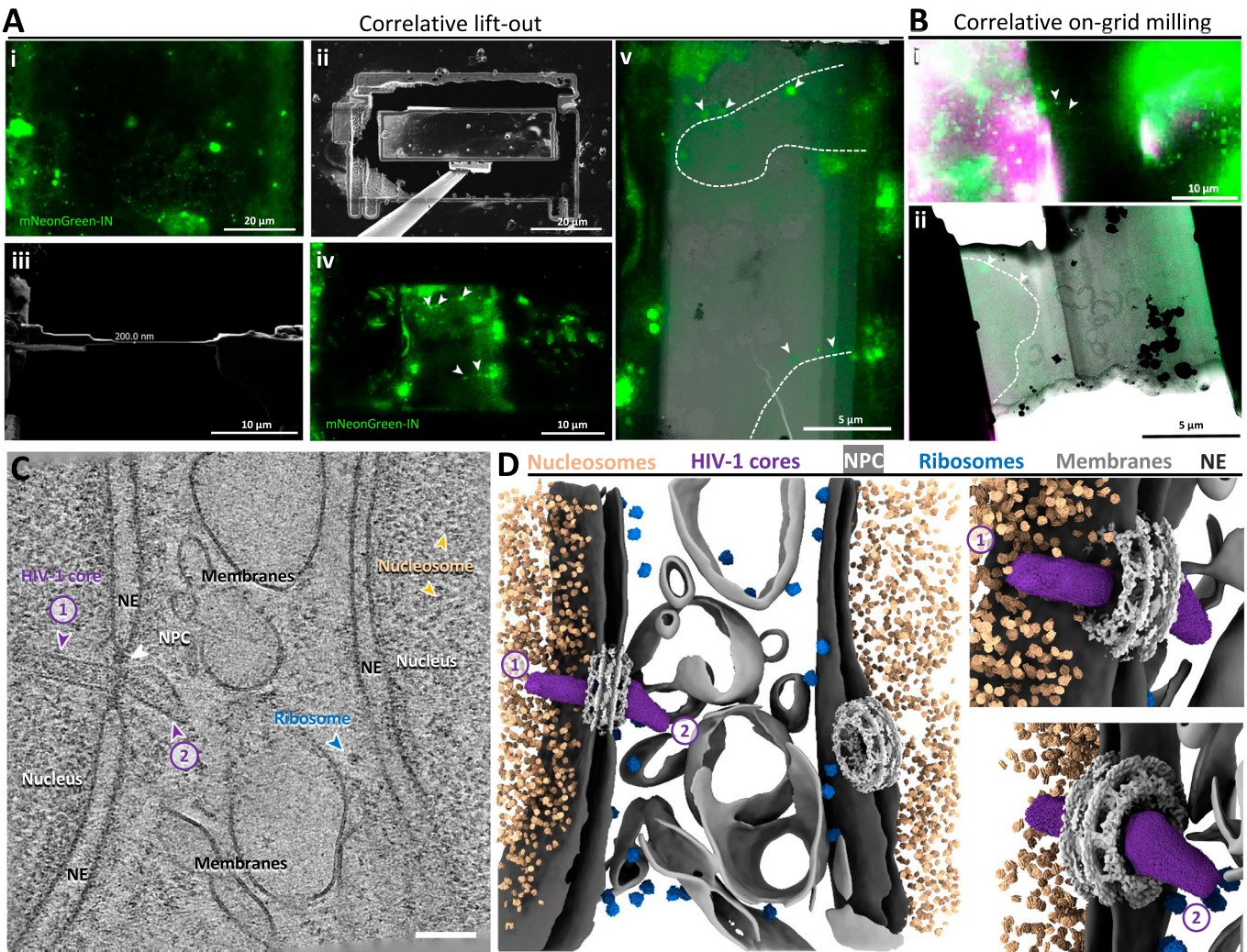

**Figure 2. Correlative cryo-FLM, cryo-FIB and cryo-ET of nuclear import of HIV-1.**

(A) Illustration of correlative planar lift-out workflow. (i) Cryo-FLM image of the targeted nucleus before lift-out, MIP from 20 images, step size = 500 nm; (ii) Cryo-FIB image of the lift-out slab with an EasyLift needle attached; (iii) Cryo-FIB image of the polished lamella; (iv) Cryo-FLM image of the polished lamella, MIP from 20 images, step size = 500 nm. (v) Cryo-FLM image overlapped with the cryo-TEM overview of the same lamella. White arrowheads point to the targeted HIV-1 cores, and white dashed lines frame the nucleus. (B) Correlative on-grid milling. 3D correlative milling was conducted in Arctis using the default pipeline or in Aquilos using the 3D correlation toolbox. (i) Fluorescence image of the polished lamella, MIP from 151 images, step size = 100 nm; (ii) Cryo-FLM image overlapped with the cryo-TEM overview of the same lamella. White arrowheads point to the targeted HIV-1 cores, and white dashed lines frame the nucleus. (C) A representative tomographic slice of a correlatively acquired tomogram. Two successive HIV-1 cores at the same NPC are indicated by purple arrowheads and numbered: No.1, an imported tube-shaped HIV-1 core; No.2, a docked cone-shaped HIV-1 core. The NPC, ribosome, and prominent nucleosomes are labelled. The nucleus, nuclear envelope (NE) and membranes are annotated accordingly. Scale bar, 100 nm. (D) The segmented volume of (C), shown as an overview (left) and zoomed-in views of the imported (upper right, No. 1) and docked (lower right, No. 2) HIV-1 cores. HIV-1 cores, NPCs, nucleosomes, ribosomes, nuclear envelope (NE) and membranes are segmented with the indicated colours. For better visualization, the NPC is illustrated by placing the published EM structure EMD-14321 on the location of pores. Source data are available online for this figure.

data showing HIV-1 accumulation at the nuclear periphery (Fig. 1A). Of the nuclear-associated cores ($n = 442$), four distinct stages of import were identified: approaching the NPC (25.3%), docking (28.5%), traversing (35.5%), and post-import within the nucleus (10.6%) (Figs. 3E and EV2A). Interestingly, the full progression of HIV-1 nuclear import was effectively captured in a single tomogram, as exemplified in Fig. 3A,B and Movie EV2. The successful combination of functional reconstitution of HIV-1 nuclear import at high abundance and efficient correlative cryo-ET targeting enabled systematic structural analysis of HIV-1 cores during nuclear import.

## Characterization of HIV-1 cores during nuclear import

To investigate how HIV-1 core morphology impacts nuclear import, we systematically analyzed core size and shapes across distinct stages of the import pathway. Of the imaged 510 HIV-1 cores, we measured 467 cores, classified based on shape (cone- vs tube-shaped) and import stage (approaching, docking, traversing, imported), excluding 43 that were partially truncated at tomogram edges.

Core width was measured at the widest point perpendicular to the long axis for statistical comparison across import states. Our analysis revealed that the average core width reduces along the

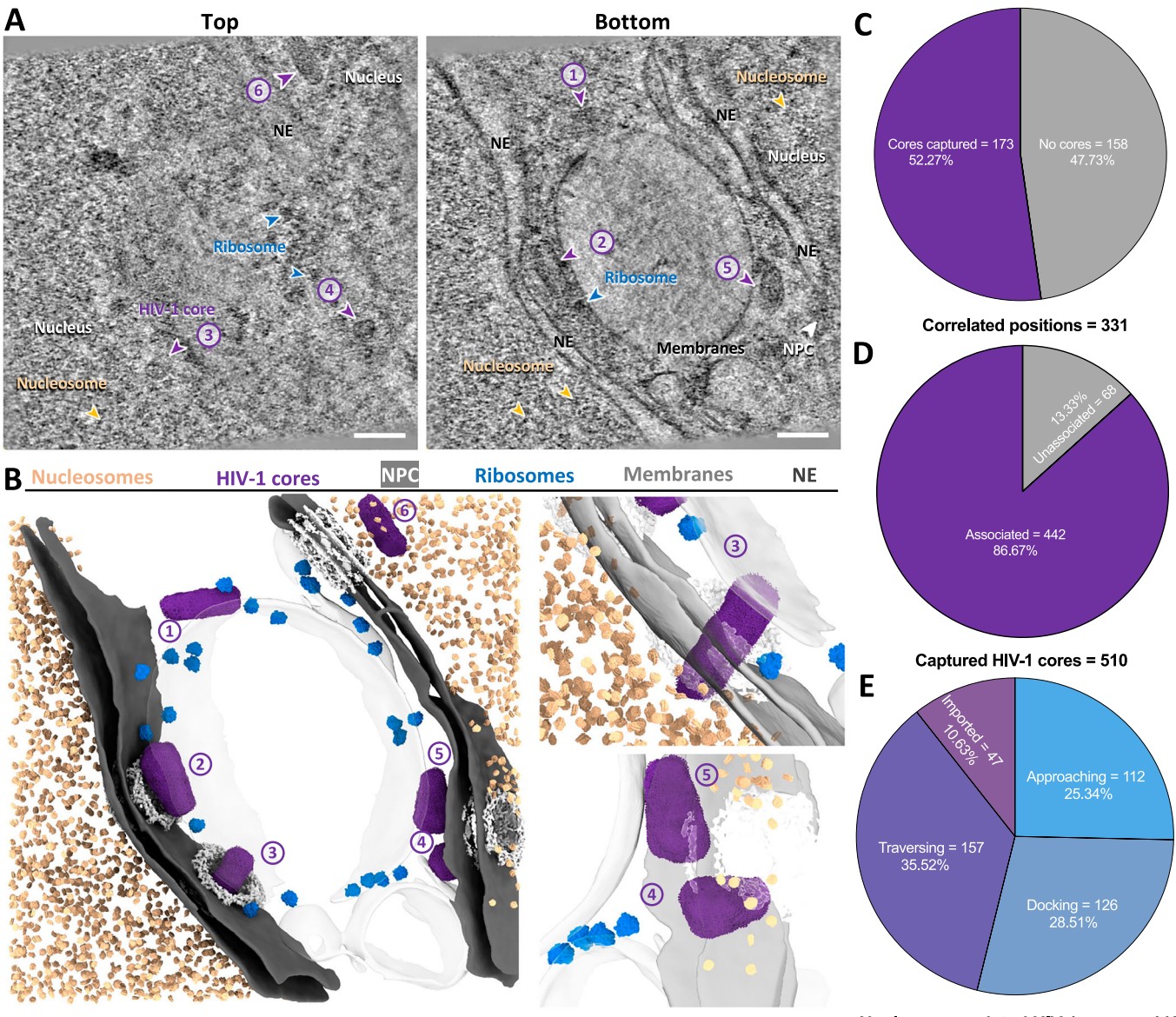

**Figure 3. Capturing HIV-1 cores in multiple states during the nuclear import.**

(A) Slices of a correlatively acquired tomogram at the top (left) and bottom (right) of the volume featuring six HIV-1 cores in multiple states. HIV-1 cores are indicated by purple arrowheads and numbered 1 through 6. The NPC, ribosome and prominent nucleosomes are labelled. The nucleus, nuclear envelope (NE) and membranes are annotated accordingly. Scale bar, 100 nm. (B) The segmented volume of (A) shown as an overview of six cores in different states (left) and zoomed-in views of the traversing (upper right) and approaching (lower right) cores. The states of cores: unassociated tube-shaped core (No. 1); docked cone-shaped core (No. 2); traversing tube-shaped core (No. 3); two approaching cone-shaped cores (Nos. 4 and 5); and imported tube-shaped core (No. 6). HIV-1 cores, NPCs, nucleosomes, ribosomes, nuclear envelop (NE) and membranes are segmented with the indicated colours as in Fig. 2. For better visualisation, the NPC is illustrated by placing the published EM structure EMD-14321 on the location of pores. (C) A pie chart illustrating the effectiveness of cryo-CLEM, where >52% of correlated fluorescence positions contain HIV-1 cores in tomograms (i.e., cores captured). (D) A pie chart illustrating the percentage of "captured" cores associated with the nucleus, including cores in the vicinity of the NE (distance ≤120 nm) and imported into the nucleus. HIV-1 cores that are distant from the NE (distance >120 nm) are regarded as 'unassociated'. (E) A pie chart showing the distribution of HIV-1 cores at each state during the nuclear import, as defined in (B). The number of cores observed in each state is indicated. Source data are available online for this figure.

import process. Notably, imported cores were ~6.2 nm narrower than docking cores, and ~1.7 nm narrower than traversing cores (Fig. 4A). The width of cores in the approaching, docking, and unassociated groups do not differ significantly, suggesting that core width is an important factor for successful translocation through the NPC.

We further explored the impact of core shape on HIV-1 nuclear import. Cone-shaped cores were measured 55.3 ± 5.7 nm wide on average at the wide-end, while tube-shaped cores were 41.4 ± 3.0 nm on average, in agreement with core dimensions measured prior to incubation (Fig. EV2B) and consistent with previously reported measurements (Briggs et al, 2003; Ni et al, 2021; Zhao et al, 2013).

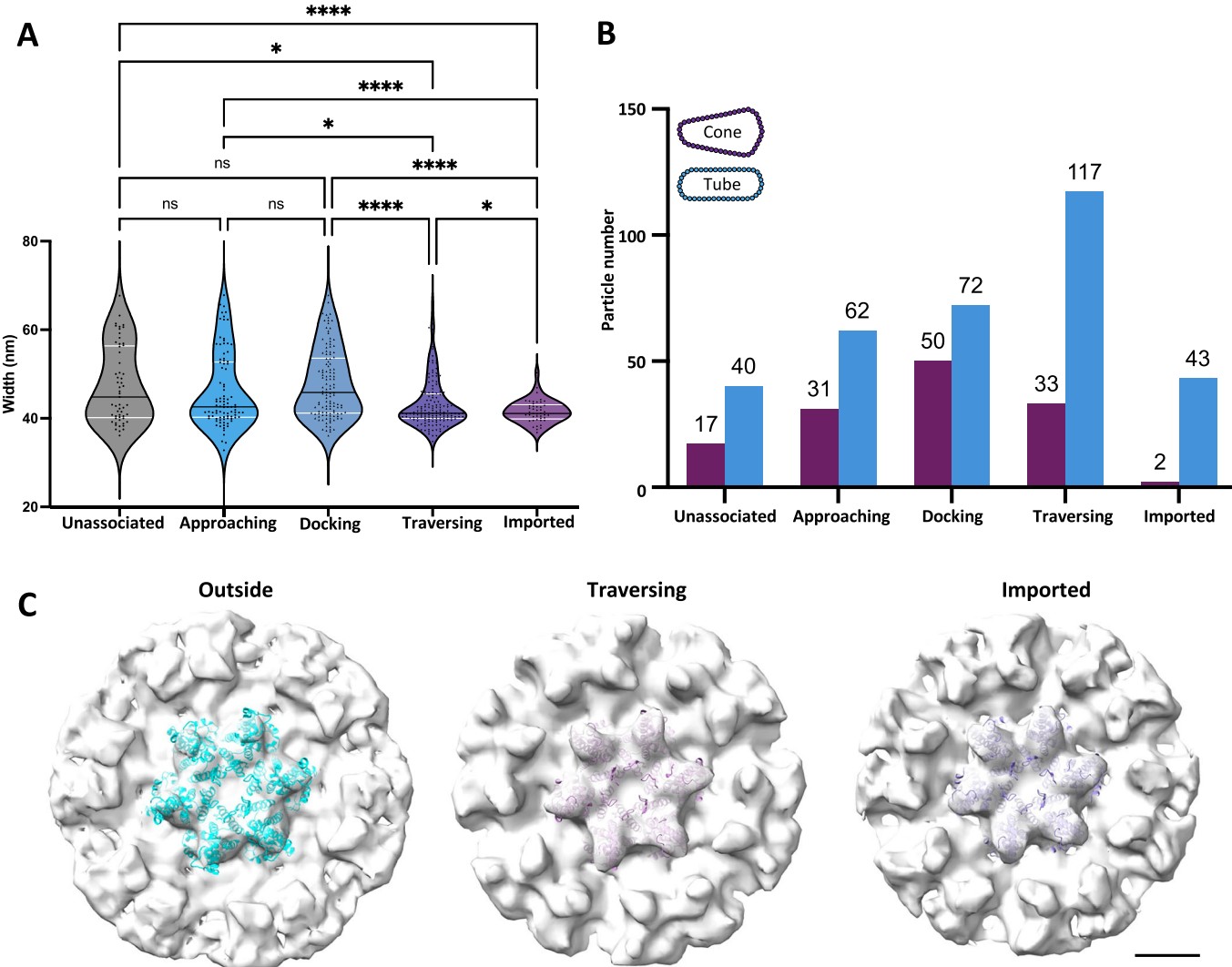

**Figure 4. Characterisation of HIV-1 cores in multiple states during the nuclear import.**

(A) A violin plot of the statistical analysis on the width of HIV-1 cores (measured at the wide end) in each state. The width of imported HIV-1 cores measures 41.56 ± 2.572 nm (SE = 0.3834, $n$ = 45), the traversing measures 43.28 ± 5.736 nm (SE = 0.4683, $n$ = 150), the docking measures 47.80 ± 7.779 nm (SE = 0.7043, $n$ = 122), the approaching measures 46.17 ± 8.453 nm (SE = 0.8765, $n$ = 93), and the unassociated measures 47.40 ± 8.615 nm (SE = 1.141, $n$ = 57). Black lines represent the medians, white lines represent the quartiles, and black dots represent individual HIV-1 cores (two-sided Brown–Forsythe and Welch ANOVA tests, * between unassociated and traversing = $P$ = 0.01282 < 0.05, * between approaching and traversing = $p$ = 0.04195 < 0.05, * between traversing and imported = $P$ = 0.04798 < 0.05, **** between unassociated and imported = $p$ = 0.0000735 < 0.0001, **** between approaching and imported = $P$ = 0.0000428 < 0.0001, **** between docking and traversing = $P$ = 0.0000024 < 0.0001, **** between docking and imported = $P$ < 1e-15, ns = no significance). (B) A bar chart illustrating the composition of HIV-1 core shapes in each state. Cone-shaped HIV-1 cores are in purple and tube-shaped cores in blue (two-sided Chi-square test, $P$ = 0.00002695 < 0.0001). (C) Structures of CA hexamers in capsid lattices of outside (approaching and docking), traversing, and imported cores determined by subtomogram averaging. Maps are aligned and contoured to the same level (3σ) and fitted with a CA hexamer model (PDB 6SKK). Scale bar, 5 nm. Source data are available online for this figure.

Since tube-shaped cores are narrower than the wide-end of cone-shaped ones, we compared the ratio of cone- versus tube-shaped cores at different import stages. While the input sample contained nearly equal number of cone- and tube-shaped cores (Fig. EV1D), 64% of cores observed outside the nucleus (unassociated, approaching, or docking) were tube-shaped, and this morphology was markedly enriched during traversal and post-import (Fig. 4B; Table EV1). Strikingly, among 45 imported cores, 95% cores were tube-shaped, with only two cone-shaped cores inside the nucleus, both notably narrower than those associated with nuclei (average width of 48.6 nm

vs. 55.3 nm). These findings suggest that NPCs act as selective filters, favoring narrower, tube-shaped HIV-1 cores. Alternatively, cone-shaped cores may fully uncoat soon after nuclear import, making them difficult to detect. We favor the former interpretation, as previous reports indicate that core uncoating occurs approximately 6 h after reaching the nucleus (Burdick et al, 2020), while our incubation time was below 1 h. However, our data cannot rule out the possibility that nuclear entry is influenced by core morphology, potentially linked to RNA content, rather than width alone, given the small number of cone-shaped cores observed inside the nucleus.

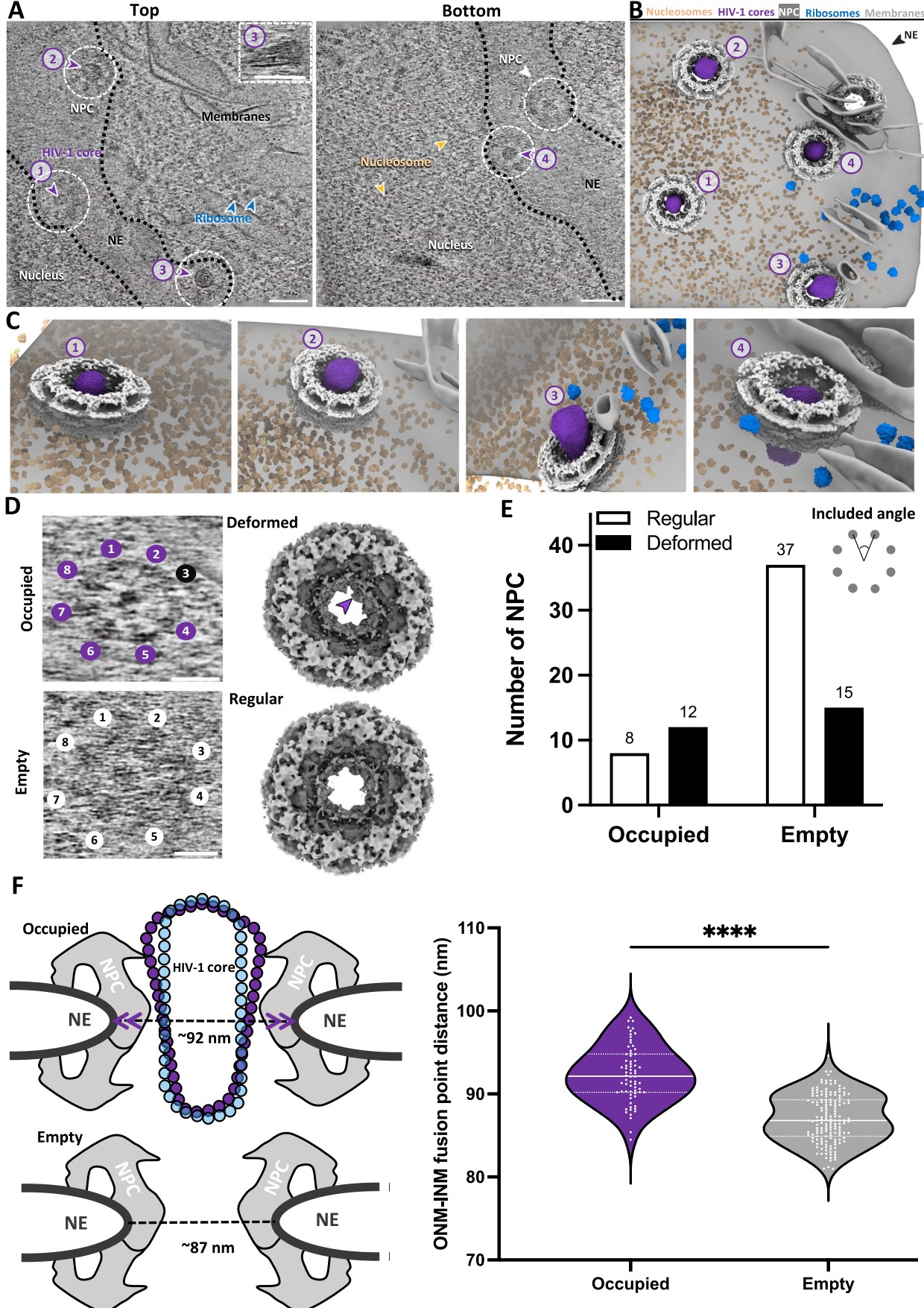

**Figure 5. Interplay between HIV-1 cores and NPCs.**

(A) Tomographic slices of a correlatively acquired tomogram at the top (left) and bottom (right) of the volume showing the face-on views of NPCs occupied by HIV-1 cores. HIV-1 cores are indicated by purple arrowheads and numbered (1–3 on the left and 4 on the right), along with one empty NPC (white arrowhead) (right). The inset depicts the side view (XZ) of core No. 3 in the NPC. NPCs are indicated by white dashed circles. The ribosomes and prominent nucleosomes, nucleus and nuclear envelope (NE) are annotated accordingly. The boundaries of NE are indicated by black dashed lines. Scale bar, 100 nm. (B, C) The segmented volume of (A) shown as an overview of four cores in NPCs (B) and zoomed-in tilted views of these four HIV-1 cores at various depths within the NPC during the traversing (C). HIV-1 cores, NPCs, nucleosomes, ribosomes and nuclear envelope (NE) are segmented with the indicated colours as in Fig. 2. For better visualisation, the NPC is illustrated by placing the published EM structure EMD-14321 on the location of pores. (D) Representative tomographic slices (left) of occupied and empty NPCs with the subunits mapped back (right). Discernible NPC subunits are numbered, offset subunit is numbered in black and indicated by the purple arrow. The mapping back uses the composite map from this study. Scale bar, 50 nm. (E) A bar chart illustrating the status of NPCs. Deformation is defined as more than one of the included angles between the subunits measure larger than 47.5° or smaller than 42.5°, deviating from the eightfold symmetric angle of 45°, n of empty NPCs = 52, n of occupied NPCs = 20 (Two-sided Fisher's exact test, $P = 0.028$). Included angle measurement is illustrated. (F) Statistical analysis of the NPC sizes with and without HIV-1 cores inside. Left, a cartoon illustrating that the size of NPCs is measured by the distance between the outer-nuclear-membrane (ONM) and inner-nuclear-membrane (INM) fusion points (indicated by dashed black lines). Purple arrowheads indicate the dilation of the NPC. Both cone-shaped and tube-shaped models of the HIV-1 core are presented. Right, a violin plot of the size distribution of NPCs with cores (purple, 92.34 ± 3.336 nm, SE = 0.3987, n = 70) and without cores (gray, 87.05 ± 2.865 nm, SE = 0.2272, n = 159). Black lines represent the medians, white lines represent the quartiles, and white dots represent individual NPCs (two-sided t test, ****$P < $1e-15). Source data are available online for this figure.

These results are consistent with prior ultrastructural observations in HIV-1-infected CD4$^+$ T cells, where tube-shaped cores account for 45%, cone-shaped cores for ~17%, and the remainder appeared partially disassembled within the nucleoplasm (Zila et al, 2021a). In contrast, HIV-1-infected macrophages exhibited predominantly cone-shaped nuclear cores, with only ~10% displaying a tubular morphology (Kreysing et al, 2025). Whether HIV-1 nuclear import selectivity is cell-type-dependent remains to be determined. It's worth noting that our work used HIV-1 cores derived from VLPs produced with the lentiviral packaging vector PsPAX2. While these cores recapitulate key structural features of mature HIV-1 cores, the absence of the full viral RNA genome may influence core properties. Notably, we observed a higher proportion of tube-shaped cores in VLPs (~50%) compared to native cores (~15%) (Fig. EV1D), although the average dimensions of cones and tubes did not differ significantly (Fig. EV1E). Whether the absence of the viral RNA genome and reverse transcription affects HIV-1 nuclear import dynamics remains to be investigated.

## Structural remodeling of NPCs during HIV-1 core translocation

The nuclear import of HIV-1 cores involves interactions of the capsid shell with NPC components, including nucleoporins such as Nup358 and Nup153 (Bichel et al, 2013; Di Nunzio et al, 2012; Lin et al, 2013; Matreyek et al, 2013; Schaller et al, 2011; Wilbourne and Zhang, 2021), as well as other host factors such as CPSF6 which has been implicated to direct the capsid to the site of integration after traversing through the NPC (Bejarano et al, 2019; Bhattacharya et al, 2014; Burdick et al, 2020; Matreyek et al, 2013; Ning et al, 2018; Price et al, 2014; Selyutina et al, 2020; Zhong et al, 2021). To investigate how capsid–NPC interactions mediate HIV-1 nuclear import and the structural integrity of the HIV-1 capsid lattice during nuclear import, we performed template matching and subtomogram averaging (STA) on the capsid lattice of HIV-1 cores at various stages: outside the nucleus (approaching and docking), traversing the NPC, and fully imported. We analyzed eleven tomograms (~120 nm thick) containing cores in the "outside" of the nucleus, eleven with traversing cores, and nine containing imported cores, all of which displayed clearly discernible CA hexameric lattices suitable for STA. The resulting CA hexamer structures were resolved at 12 Å (outside), 13.5 Å (traversing), and 13.5 Å (imported) resolution (Fig. EV2D). The STA maps of CA

hexamers fit well with the CA hexamer model (PDB 6SKK) derived from our previous CA tubular assemblies (Fig. 4C) (Ni et al, 2020). Across all stages, we observed no substantial conformational changes in the hexamer structure.

To investigate potential structural adaptations of both the HIV-1 capsid and the NPC during nuclear import, we focused our analysis on cores captured in the traversing state (i.e., positioned within the NPC channel) as exemplified in Fig. 5A–C for a top view where different degrees of penetration of HIV-1 cores into NPCs can be delineated (Fig. 5A–C; Movie EV3). We mapped the STA-refined coordinates of individual CA hexamers back onto the original tomograms to assess capsid integrity and found that most cores inside the NPCs appeared to have a complete lattice. Previous atomic force microscopy and molecular dynamics studies suggest that capsid elasticity is critical for successful nuclear entry (Deshpande et al, 2024; Hudait and Voth, 2024; Rankovic et al, 2021). However, the dynamic aspects of capsid remodeling during nuclear import remain unclear. Complementary approaches such as super-resolution fluorescence microscopy, particularly MIN-FLUX, combined with direct capsid labeling using ncAA incorporation and click chemistry, will be critical for probing the dynamical interplay between the HIV-1 core and the NPC (Deguchi et al, 2023; Gwosch et al, 2020; Schifferdecker et al, 2022).

To evaluate whether the NPC itself undergoes structural changes during core translocation, we assessed the eightfold rotational symmetry by template matching, followed by subtomogram refinement and averaging (Fig. EV2C,D). We observed disruptions in the eightfold symmetry of the cytoplasmic ring in a subset of the core-occupied NPCs (Fig. 5D, top) while NPCs without a core inside consistently retained canonical symmetry (Fig. 5D, bottom). To quantify potential structural changes in the NPC, we calculated the included angles between subunits and measured the diameters of NPCs, defined as the fusion points of the outer and inner nuclear membranes. NPCs occupied by HIV-1 cores exhibited substantial deformation while compared to empty NPCs (Fig. 5E). Moreover, occupied NPCs showed a significant expansion compared to empty NPCs, increasing from an average of 87 nm to 92 nm, reflecting an approximate 5.5 nm dilation (Fig. 5F). Together, these data suggest that HIV-1 nuclear import triggers NPC remodeling. This highlights the structural elasticity of NPCs, which are known to adapt their conformation in response to cellular state (Mahamid et al, 2016; Mosalaganti et al, 2022; Schuller et al, 2021; Singh et al, 2024; Zimmerli et al, 2021).

Notably, the diameter of nuclear pore complexes (NPCs) measured at the inner/outer nuclear membrane (INM/ONM) fusion point in permeabilized T cells was ~87 nm. This is significantly larger than the ~79 nm observed in isolated HeLa nuclear envelopes (von Appen et al, 2015), but slightly smaller than the 92–96 nm range reported in cryo-FIB-milled intact human cells (Mosalaganti et al, 2022; Schuller et al, 2021; Zila et al, 2021a). While NPC diameter may vary depending on cell type and physiological state, NPCs in permeabilized T cells may exhibit partial constriction due to preparation-induced stress. Nonetheless, numerous nuclear import events were captured, with HIV-1 cores observed translocating through NPCs. Further optimization of the CEM nuclear preparation protocol, such as incorporating an energy regeneration system (Hou et al, 2025; Zimmerli et al, 2021), will be important to better preserve native NPC architecture and functionality. Additionally, whether the impact of HIV-1 infection on NPC structure is cell-type-dependent remains to be determined. For example, infection in CD4+ T cells did not appear to cause NPC deformation (Zila et al, 2021a), whereas in macrophages, HIV-1 cores were reported to "crack" the NPC during nuclear entry, based on analysis of 10 traversing cores and their corresponding NPCs, supported by molecular dynamics simulations (Kreysing et al, 2025).

In summary, we present a robust workflow for the in situ structural characterization of HIV-1 nuclear import. This approach enables quantitative analysis of HIV-1 core–NPC interactions and lays the groundwork for future investigations into the effects of HIV-1 capsid mutants and host factors on nuclear import. Furthermore, STA can be employed to directly visualize capsid–NPC interactions, offering molecular insights into the complex interplay between the HIV-1 capsid and the NPC under near-native conditions (Bichel et al, 2013; Di Nunzio et al, 2012; Fu et al, 2024; Lin et al, 2013; Matreyek and Engelman, 2011; Matreyek et al, 2013; Schaller et al, 2011). Ultimately, our system can be leveraged to study nuclear import mechanisms of other viruses, such as herpes simplex virus, adenoviruses, and hepatitis B virus (Cohen et al, 2011).

# Methods

### Reagents and tools table

| Reagent/resource | Reference or source | Identifier or catalog number |
| --- | --- | --- |
| **Experimental models** | | |
| Human embryonic kidney (HEK) 293T Lenti-X cells (*H. sapiens*) | Takara/Clontech | Cat: 632180 |
| CD4 + T lymphocyte CEM cells (*H. sapiens*) | NIH HIV reagent program | ARP-117 |
| HEK293T cells (*H. sapiens*) | ATCC | CRL-11268 |
| **Recombinant DNA** | | |
| pNL4-3 (Env-) vectors | Adachi et al, 1986 | n/a |
| NL4-3 Env expression vector pIIINL4env | Murakami and Freed, 2000 | n/a |
| psPAX2 | Addgene | Plasmid #12260 |
| VPR-mNeongreen-IN | Gift from Prof. Zandrea Ambrose's laboratory at the University of Pittsburgh) | n/a |
| **Antibodies** | | |
| Anti-Nup358 antibody | Abcam | ab64276 |

| Reagent/resource | Reference or source | Identifier or catalog number |
| --- | --- | --- |
| Goat anti-rabbit Alexa 568 | Abcam | ab175471 |
| **Oligonucleotides and other sequence-based reagents** | | |
| **Chemicals, enzymes and other reagents** | | |
| DMEM | Gibco | 11965092 |
| Fetal bovine serum (FBS) | ThermoFisher | 10270106 |
| L-glutamine | Gibco | 25030081 |
| MEM non-essential amino acids | Gibco | 11140050 |
| RPMI-1640 | Sigma-Aldrich | R0883 |
| GenJet II | SignaGen Laboratories | SL100488 |
| Lipofectamine 2000 | Invitrogen | 11668019 |
| OptiPrep | Sigma-Aldrich | D1556 |
| Digitonin, high purity | Sigma-Aldrich | 300410 |
| Triton X-100 | Sigma-Aldrich | T8787 |
| Ethylene glycol bis succinimidyl succinate (EGS) | ThermoFisher | 21565 |
| **Software** | | |
| FIJI/ImageJ | Schindelin et al, 2012 | https://imagej.net/software/fiji/ |
| Leica Application Suite X (Leica) version 3.5.9.26787 | Leica | https://www.leica-microsystems.com/products/microscope-software/p/leica-las-x-ls/ |
| WebUI version 1.1 | Spurný et al, 2023 | |
| AutoTEM 5 version 5.19 | ThermoFisher | https://www.thermofisher.com/uk/en/home/electron-microscopy/products/software-em-3d-vis/autotem-5-software.html |
| METEOR version 1.0 | Delmic | https://www.delmic.com/en/products/cryo-solutions/meteor |
| iFLM system version 1.3 | Yang et al, 2023 | |
| Tomography 5 version 5 | ThermoFisher | https://www.thermofisher.com/uk/en/home/electron-microscopy/products/software-em-3d-vis/tomography-software.html |
| EPU version 3.8 | ThermoFisher | https://www.thermofisher.com/uk/en/home/electron-microscopy/products/software-em-3d-vis/epu-software.html |
| FEI TIA version 0.7.1 | ThermoFisher | https://fei-tia.software.informer.com/download/ |
| MotionCorr2 version 1.4.0 | Zheng et al, 2017 | |
| IMOD version 4.11.1 | | |
| Prism 10 | GraphPad | https://www.graphpad.com |
| emClarity version 1.5.0.2 | Himes et al, 2018 | |
| emClarity version 1.5.3.10 | Ni et al, 2022 | |
| RELION version 4.0 | Kimanius et al, 2021 | |
| UCSF ChimeraX version 1.9 | Goddard et al, 2018 | https://www.cgl.ucsf.edu/chimerax/ |
| IsoNet version 0.2 | Liu et al, 2022 | |

| Reagent/resource | Reference or source | Identifier or catalog number |
|---|---|---|
| MemBrainseg version 0.0.8 | Isensee et al, 2021 | |
| Amira version 2024.2 | ThermoFisher | https://www.thermo fisher.com/nl/en/home/electron-microscopy/products/software-em-3d-vis/amira-software.html |
| ArtiaX version 0.6.0 | Ermel et al, 2022 | https://github.com/FrangakisLab/ArtiaX |
| MagpiEM | | https://github.com/fnight128/MagpiEM |
| **Other** | | |
| Dounce homogenizer | Kimble | 885300-0040 |
| SW32Ti rotor and SW41Ti rotor | Beckman | n/a |
| Beckman Coulter Optima XPN-90 | Beckman | n/a |
| Leica TCS SP8 confocal microscope | Leica | n/a |
| Leica EM GP2 automated plunge freezer | Leica | n/a |
| Glacios | ThermoFisher | n/a |
| Arctis | ThermoFisher | n/a |
| Aquilos 2 | ThermoFisher | n/a |
| Krios4 | ThermoFisher | n/a |

## Mammalian cell culture

Cell lines were maintained in an incubator at 37 °C and 5% $CO_2$. Human embryonic kidney (HEK) 293T Lenti-X cells (Takara/Clontech 632180, authenticated by the vendor using STR profiling) cells were cultured in DMEM (Gibco) (Sigma-Aldrich) medium supplemented with 10% FBS, 2 mM L-glutamine (Gibco) and 1% MEM non-essential amino acids (Gibco). CD4+ T lymphocyte CEM cells (NIH HIV reagent program/ARP-117) were cultured in RPMI-1640 (Sigma-Aldrich) medium supplemented with 10% FBS and 2 mM L-glutamine (Gibco). CD4+ T lymphocyte CEM cells (ARP-117; NIH HIV Reagent Program) have not been authenticated.

## HIV-1 virion production

To produce HIV-1 virus particles, HEK293T cells (ATCC, CRL-11268, authenticated by the vendor using STR profiling) were transfected with pNL4-3 (Env-) vectors together with NL4-3 Env expression vector pIIINL4env using Lipofectamine 2000 (Invitrogen). Culture media from transfected cells were harvested at 48 h post-transfection and cleared by filtration through a 0.45 µm polyvinylidene fluoride (PVDF) filter. The virions were concentrated by ultracentrifugation through an 8% OptiPrep density gradient (Sigma-Aldrich) ($100,000 \times g$, Sorvall AH-629 rotor) for 1 h at 4 °C. The concentrated virions were further purified by ultracentrifugation ($120,000 \times g$, Sorvall TH-660 rotor) through a 10–30% OptiPrep gradient for 2.5 h at 4 °C. The opalescent band was harvested, diluted with PBS, and ultracentrifuged at $110,000 \times g$ at 4 °C for 2 h. Pelleted particles were resuspended in 5% sucrose/PBS solution and stored at −80 °C until use.

## T cell plasma membrane permeabilization

T cell plasma membrane permeabilization was performed with cells in exponential growth phase, at a concentration of ~$1 \times 10^6$/ml. Cells were directly pelleted by centrifugation for 5 min, $500 \times g$ at 4 °C. Cell pellets were resuspended in 10 ml ice-cold PBS and all subsequent steps were performed on ice. The cells were pelleted by centrifugation for 5 min, $500 \times g$ at 4 °C, and the PBS wash was repeated. Cell lysis was performed using either mechanical lysis or detergent treatment. For mechanical lysis, the washed cell pellet was resuspended in hypotonic buffer (50 mM Tris-HCl pH 7.5 or 50 mM HEPES-NaOH pH 7.5 and protease inhibitor, EDTA-free (Roche/Pierce)) and incubated for 40 min on ice. The cells were mechanically lysed by 15–40 passes up and down using either a 40 ml Dounce homogenizer or a 2 ml Dounce homogenizer with tight pestle B, clearance 0.0010–0.0030 in/0.025–0.076 mm (40 ml) (Kimble 885300-0040) or clearance 0.0005–0.0025 in/0.0127–0.0635 mm (2 ml) (Kimble 885300-0002). For detergent treatment, the washed cells were incubated in the hypotonic buffer supplemented with 0.025% digitonin at room temperature with rotation for 10 min (Ribbeck and Gorlich, 2001). KCl and $MgCl_2$ were added to the lysates to get final concentrations of 25 mM KCL and 5 mM $MgCl_2$ and incubated on ice for 20 min.

## HIV-1 virus-like particle production and core isolation

The protocol was adopted from (Aiken, 2009) with adjustments. All sucrose buffers are filtered using a 0.20-µm filter (Sarstedt). In all, 1× STE (10 mM Tris-HCl pH 7.4 + 100 mM NaCl + 1 mM EDTA pH 8.0) and 1 mM inositol hexakisphosphate (IP6) was added fresh to the buffers. HIV-1 VLPs were produced by transfecting Lenti-X HEK293T cells at ~80% confluence using GenJet II (SignaGen Laboratories). Each transfection was carried out in eight T75 flasks. For each prep, 40 µg of psPAX2 (Addgene # 12260) and 10 µg of pVPR-mNeongreen-IN plasmid DNA (a gift from Prof. Zandrea Ambrose's laboratory at the University of Pittsburgh) was mixed with 2 ml DMEM without FBS. 100 µl of GenJet II mixed with 2 ml DMEM without FBS was added to the plasmid DNA mix and incubated for 10 min at room temperature (RT). Media of the flasks was refreshed with 7.5 ml/ flask. Overall, 500 µl transfection mix per flask was added dropwise and incubated for 48 h.

The medium was harvested, and cells were pelleted by centrifugation for 5 min, 220 g at 20 °C. The supernatant was filtered using a 0.45-µm filter (Sarstedt). The filtered supernatant was added to 38.5 ml ultracentrifuge tubes (Beckman Coulter 344058) and underlaid with 5 ml 20% sucrose 1x STE (10 mM Tris-HCl pH 7.4, 100 mM NaCl, 1 mM EDTA pH 8.0) + 1 mM IP6 sucrose cushion, using a 5 ml stripette. The VLPs were pelleted by centrifugation using the SW32Ti rotor (Beckman Coulter Optima XPN-90) for 3 h, 30 k RPM at 4 °C. After centrifugation, the supernatant and sucrose cushion were removed, and the tube was placed inverted on a tissue to dry for ~5 min. The remaining liquid was removed by cleaning the inside walls of the tube using a tissue (Kimtech) without touching the pellet. Each VLP pellet was resuspended in 400 µl 1× STE supplemented with 1 mM IP6.

A sucrose gradient was prepared in a 13.2 ml ultracentrifuge tube (Beckman Coulter 344059). All buffers were supplemented with 1× STE and 1 mM IP6. The layers were added from bottom to top using 2 ml stripettes: 2 ml 85% sucrose, 1.7 ml 70% sucrose, 1.7 ml 60% sucrose, 1.7 ml 50% sucrose, 1.7 ml 40% sucrose and 1.7 ml 30% sucrose and placed in the fridge for ~7–8 h. Before use, 250 µl 15%

sucrose supplemented with 1% Triton X-100 was added on top of the gradient and a layer of 7.5% sucrose was added on top of the Triton layer to provide a barrier between the VLPs and the Triton before centrifugation. Finally, the 800 μl concentrated VLPs were added on top of the 7.5% sucrose layer and topped off with 1× STE 1 mM IP6. The sample was separated by centrifugation using a SW41Ti rotor (Beckman Coulter Optima XPN-90) for 15–17 h, 33k RPM at 4 °C.

The sample was harvested immediately after stopping the centrifuge. The mNeonGreen band corresponding to HIV-1 cores was visualized using a blue light transilluminators and collected by side puncturing the ultracentrifugation tube using a 25 G needle (BD Microlance 3). The isolated cores were either snap frozen in single-use aliquots with liquid nitrogen and stored at −80 °C or dialyzed overnight at 4 °C against 500 ml of 1× SHE buffer (10 mM HEPES-NaOH pH 7.4 + 100 mM NaCl + 1 mM EDTA pH 8.0) supplemented with 1 mM IP6 using a 0.1–0.5 ml 7 kDa cut-off Slide-A-Lyzer Dialysis Cassette (Thermo Fisher Scientific) before mixing with permeabilized T cells. HIV-1 core preparations were repeated multiple times, consistently yielding cores of similar quality assessed by cryo-EM imaging (methods below).

## Mixing HIV-1 cores with permeabilized T cells

The nuclei of permeabilized T cells were stained with 1 μM SiR-DNA (Spirochrome) on ice for 1 h. The HIV-1 cores were mixed at an identical volume ratio with $6 \times 10^6$/ ml mechanically isolated nuclei or permeabilized T cells and incubated for 30 min on ice, followed by incubation for 30 min at 37 °C. For cryo-FIB and cryo-ET, the incubated samples were then fixed with ethylene glycol bis succinimidyl succinate (EGS). EGS fixation was performed with 5 mM for 30 min at RT followed by inactivation using 50 mM Tris-HCl pH 7.5 and placed back on ice.

## NUP358 antibody labeling

EGS fixed permeabilized T cells mixed with HIV-1 cores were added to poly-D-lysine (50 μg/ml) (Gibco) coated Ibidi μ-slides (Ibidi) by centrifugation at $100 \times g$ for 5 min at 4 °C. Wells were washed 1× with PBS and incubated with PBS + 1% BSA for 20 min at RT. After blocking, the sample was incubated overnight at 4 °C with anti-Nup358 antibody (Abcam ab64276) at 1:2000 in PBS + 1% BSA. The sample was washed 3× with PBS and incubated with goat anti-rabbit Alexa 568 (ab175471) at 1:1000 in PBS + 1% BSA for 45 min at RT. Finally, the sample was washed 3× in PBS and used for imaging the same day.

## Confocal microscopy, live imaging and fluorescence image processing

For all light microscopy imaging, the Leica TCS SP8 confocal microscope equipped with a HC PL Apo x63 MotCORR Water CS2 Objective NA 1.2 and a HyD GaAsP detector, controlled with LAS X software was used (Leica). Excitation with 488 nm, 561 nm and 633 nm laser lines were used and dynamic filter settings were applied. Z-stacks were collected with a 0.3–0.5 μm step size. For both imaging and Z-stacks, a 1 AU pinhole and a 1024 × 1024 resolution was used. Images, Z-stacks, were processed using Leica Application Suite X (Leica) and FIJI ImageJ (Schindelin et al, 2012).

For counting the mNeonGreen-IN puncta decorating the nuclei, the 3D objects counter plugin in Fiji ImageJ was used (Bolte and Cordelieres, 2006). Data from two biological replicates for mechanical lysis and two biological replicates for digitonin permeabilization were pooled and used for counting. The nuclei volumes were segmented in Fiji ImageJ (Schindelin et al, 2012) prior to counting to avoid false positive NG-IN signal from the background. A size threshold of 7–27 was used depending on the pixel size of the image. An intensity threshold ranging from 20 to 110 (8-bit, 255 scale) was used. Counts for every individual nucleus were inspected manually to confirm reliable and accurate counting of the software. Sample sizes were selected in alignment with those used in previously published fluorescence imaging studies on HIV-1 nuclear import (Li et al, 2021).

## Plunge freezing vitrification

The samples were plunge frozen on glow-discharged gold finder grids, R2/2 Au G300F1 (Quantifoil), or copper grids, R2/1 Cu 300 (Quantifoil), using the Leica EM GP2 automated plunge freezer (Leica). The sample preparation was conducted in the blotting chamber at 20 °C with 95% humidity. Experimental samples were incubated with 10% glycerol for 2 min prior to plunge freezing. A mixture of 3–6 μl of HIV-1 cores and mechanically isolated nuclei or permeabilized T cells was added to the carbon side of the grid, while 2–4 μl of 1× STE + 1 mM IP6 or PBS was added to the back side of the grid. The grids were back blotted for 6 s using filter blotting paper (Whatman) and immediately plunge frozen in liquid ethane. For control groups, 3.5 μl of full-genome virions and cores from VLPs were added to the carbon side of the grid, blotted from the back for 3 s, and then plunge frozen in liquid ethane. Following vitrification, grids were clipped and initially screened on a Glacios cryo-electron microscope (Thermo Fisher Scientific). Grids exhibiting optimal ice quality and particle distribution were selected for subsequent data collection.

## Correlative cryo-FIB milling

The vitrified mixture of HIV-1 cores with either mechanically isolated nuclei or permeabilized T cells, was further thinned by cryo-FIB milling to prepare lamellae, guided by cryo-CLEM in two systems. Five grids were then loaded by a robotic delivery device (Autoloader) onto a dual-beam FIB/SEM microscope, Arctis (Thermo Fisher Scientific). This microscope is equipped with a cryogenic stage cooled to −191 °C, a wide-field integrated fluorescence microscope (iFLM) system (Yang et al, 2023). with a ×100 objective (NA 0.75), and a plasma multi-ion source (argon, nitrogen, xenon, and oxygen), with argon used as the FIB source in this study.

Before milling, an organometallic platinum layer was deposited on the grid using the GIS system (Thermo Fisher Scientific) for 50 s. The 3D correlative milling was performed using the embedded protocol in WebUI version 1.1 (Thermo Fisher Scientific), with the milling angle adjusted to 10°. For each position, a 15-μm stack composed of fluorescence (GFP) and reflection images was acquired at a step size of 500 nm after rough milling using default parameters. The positions of targeted fluorescence spots were calculated using discernible ice chunks as fiducial markers in both SEM and FIB images, guiding the placement of lamella preparation patterns. The lamellae were produced in a stepwise sequence: (i) opening at 2 nA, (ii) rough milling at 0.74 and 0.2 nA, and (iii)

polishing at 60 pA, with the final thickness of lamellae set to 140 nm. To enhance signal detection on polished lamellae, the step size was changed to 100 nm, resulting in a 15-µm stack composed of 101 images in both GFP and far-red channels.

Six grids were loaded onto a dual-beam FIB/SEM microscope, Aquilos 2 (Thermo Fisher Scientific), equipped with a cryo-transfer system and rotatable cryo-stage cooled to −191 °C by an open nitrogen circuit. The Aquilos 2 FIB/SEM microscope was modified to accommodate a FLM system, METEOR, with a ×50 objective (NA 0.8) (Delmic Cryo BV). Grids were mounted on a METEOR shuttle with a pre-tilt of 26°, followed by coating with an organometallic platinum layer using the GIS system (Thermo Fisher Scientific) for 30 s. The milling angle was set to 10°. Permeabilized T cells and mechanically isolated nuclei seeded in optimal positions (near the center of the grid square) were selected for lamella production. Before milling, fluorescence stacks were collected for all selected positions in both GFP and far-red channels at a step size of 200 nm, ranging ±6 µm from the focal point. The stacks were further processed in ImageJ (Schneider et al, 2012) to enhance the signal-to-noise ratio (SNR). FIB and SEM images were then acquired and correlated with the fluorescence images using the open-source software 3D Correlation Toolbox (Arnold et al, 2016), with discernible ice chunks used as fiducial markers. Lamella preparation patterns were then placed based on the correlated positions, followed by sequential milling conducted by the software AutoTEM 5 (Thermo Fisher Scientific) from 0.5 nA (rough milling) to 0.3 nA (medium milling), 0.1 nA (fine milling), 60 pA (first polishing), and 30 pA (final polishing), with the final thickness of lamellae set to 120 nm. After final polishing, fluorescence stacks of lamellae were acquired in both GFP and far-red channels at a step size of 100 nm, ranging ±2 µm from the focal point. The light intensity and exposure time were set to 400 mW and 300 ms for the METEOR system. In total, 35 lamellae and 85 lamellae with discernible fluorescence were produced in Arctis and Aquilos 2, respectively.

## Correlative planar lift-out

To perform the lift-out, a rectangular 400 × 100 mesh TEM support grid (Agar Scientific) was loaded as the receiving grid, along with the sample grid, on a 35° Aquilos 2 AutoGrid (AG) shuttle (Thermo Fisher Scientific). Before milling the bulk sample, a copper companion block (10 µm × 8 µm × 5 µm) was created from a grid bar of the receiving grid and attached to the EasyLift needle (Thermo Fisher Scientific) using FIB redeposition methods. A fluorescence overview was acquired using the iFLM system (Thermo Fisher Scientific) with a ×20 objective in both GFP and far-red channels. The intensity was set to 20% and the exposure time to 200 ms. Feasible positions, or regions of interest (ROIs), with sufficient fluorescence were selected for the bulk milling.

Planar lift-out of each ROI included four main steps. These were bulk milling to create a chunk from the ROI, lift-out of the chunk, attachment of the chunk to the receiver grid, and cleaning of the front edge of the chunk (the edge that would be perpendicular to the FIB during milling to target thicknesses). A fifth step with the GIS was performed after all chunks were attached to the receiver grid. A 30 kV FIB was used in all steps. To achieve near parallel FIB milling angles relative to the surface of the bulk sample, and allow for feasible milling angles of a chunk on the receiver grid, the following formula was used to determine adequate stage tilt angles

during the lift-out and attachment steps:

$$FT = (LO - A) - (38 - PT)$$

where FT is the stage tilt angle for FIB milling of a chunk, LO is the stage tilt angle during lift-out, A is the stage tilt angle during attachment to the receiver grid, 38 is the angle between the stage and FIB, and PT is the shuttle pre-tilt angle. In this planar lift-out case, FT was the stage tilt angle at −70° stage rotation (during milling of the chunks). LO and A were the stage tilt angles when the stage rotation was 110° (during the lift-out and attachment steps).

To perform the bulk milling, the stage was readjusted with a rotational angle of 110° and a stage tilt of 17° (at this position, the sample surface was perpendicular to the FIB). Bulk milling was then performed around each ROI using FIB milling currents from 1-7 nA, producing a fluorescence chunk (55 µm × 20 µm × 4 µm). High currents were used for milling thick regions, such as near grid bars, and lower currents were used for thinner regions and cleaning the edges of an ROI. Material was removed from the top, bottom, and sides of each ROI with tabs left preserved until the lift-out steps.

After bulk milling, the chunk was prepared for lift-out with the companion block on the EasyLift. The FIB was used to prepare a clean landing area on the chunk and on the block, which ensured maximum surface area contact. Once the block was contacting the chunk, attachment was achieved with 50 pA FIB current using 0.3 µm × 0.6 µm cleaning cross-section (CCS) patterns with 800–1000 ms dwell times and 2 passes. Each pattern was run for 10 s. Patterns were positioned side-by-side in groups of three and arranged along the edge of the companion block where the milling direction was from the sample towards the block. The beginning of each pattern was positioned at the point of contact between the block and chunk. After attachment, the tabs were milled, and the chunk was lifted from the bulk sample.

To perform attachment of the chunk to the receiver grid, the stage tilt was set to 9°. The FIB was used to create a landing area on the parallel grid bars and clean the edges of the chunk. The chunk was positioned between the bars and attached on the left and right sides. Attachments between the chunk and the receiver grid bars was achieved using 300 pA FIB current with 3 µm × 1 µm × 3 µm CCS patterns. The milling direction of the patterns was set towards the grid bar. Once the chunk was attached to the grid, the FIB was used to cut the companion block free of the chunk. To re-enforce the attachment of the chunk to grid, the stage tilt was set to 17° and more attachment points were created.

With the stage tilt set to 17°, the chunk was ready for the fourth step of the lift-out, which was to clean the front edge that would face the FIB during milling. After each chunk was cleaned and ready for FIB milling, the fifth and final step of the lift-out was to rotate the stage to −70°, and tilt to 5°. At this position, a 90 s GIS coating was used to condense a protective layer on the cleaned front edges of all the attached chunks on the receiver grid.

At the −70° rotation and 5° stage tilt (8° milling angle with the 35° pre-tilt shuttle), milling then commenced, following the stepwise protocol described in the previous cryo-FIB milling section, guided by the remaining fluorescence. In total, five lamellae with discernible fluorescence were produced from the planar lift-out in Aquilos 2.

## Cryo-electron tomography data collection and data processing

For the experimental group, lamellae were transferred to three FEI Titan Krios G3 (Thermo Fisher Scientific) electron microscopes operated at 300 kV and equipped with a Falcon 4i detector and a Selectris X energy filter (Thermo Fisher Scientific). Objective apertures of 100 μm were inserted. Lamella overviews were generated by stitching images acquired at a magnification of 8100×. The TEM lamella overviews were then correlated with the fluorescence lamella images in ImageJ (Schneider et al, 2012). Following that, tilt series were collected on the correlated sites and neighboring areas without overlap. The collection was performed using Tomography 5 software (Thermo Fisher Scientific) at a magnification of 64k.

A total of 97 tilt series were collected with a nominal physical pixel size of 1.903 Å/pixel, 167 tilt series with a nominal physical pixel size of 1.94 Å/pixel, and 5 tilt series from the lift-out lamellae with a nominal physical pixel size of 2 Å/pixel. For tilt series from correlatively milled lamellae, the defocus value was set from −3 to −5 μm. The pre-tilts of lamellae were determined at ±10° or ±14°, and a dose-symmetric scheme was applied, ranging from −42° to +62° or −38° to +66° with an increment of 2°. A total of 53 projection images with 10 movie frames each were collected for each tilt series, with the dose rate set to 3 e/Å²/tilt, resulting in a total dose of 159 e/Å². For tilt series from lift-out lamellae, the defocus was set from −2 to −3 μm, with no pre-tilt set. A dose-symmetric scheme was applied with a tilt range of ±60° from 0° with an increment of 2°. Tilt series were collected at a dose rate of 2.5 e/Å²/tilt in EER format, leading to a total dose of 152.5 e/Å².

For the control groups, grids were loaded onto a FEI Titan Krios G2 (Thermo Fisher Scientific) electron microscope operated at 300 kV and equipped with a Gatan BioQuantum energy filter and post-GIF K3 detector (Gatan, Pleasanton, CA). A 100 μm objective aperture was inserted. For the grid of full-genome virions, 58 tilt series were collected from a representative cryo-EM grid using Tomography 5 software (Thermo Fisher Scientific) at a magnification of 81k with a nominal pixel size of 1.34 Å/pixel. A dose-symmetric scheme was applied with a tilt range of ±60° from 0° with an increment of 3°. Defocus was set from −1.5 to −3 μm, and the dose rate was set to 3 e/Å²/tilt. Forty-one projection images with 10 movie frames each were collected for each tilt series, resulting in a total dose of 123 e/Å². In parallel, micrographs of VLP cores were collected from a representative cryo-EM grid using EPU (Thermo Fisher Scientific) at a magnification of 105k with a nominal pixel size of 0.831 Å/pixel. A total of 59 frames were collected for each micrograph with a total dose of 22 e/Å², and the defocus was set from −3.0 to −4.0 μm. Super-resolution mode was employed, and 2000 micrographs were collected. Overall data statistics and experimental parameters for cryo-FIB lamella preparation and cryo-ET data collection and subtomogram averaging are detailed in Tables EV2 and 3. For clarification, we only correlated the signals on the periphery and inside of nuclei on the lamellae, which were used to position the target areas in the TEM overview for cryo-ET data collection. There were several reasons for signals which could not be correlated, including: (1) The mNeongreen-labeled IN had departed from capsid; (2) The autofluorescence signals that were not associated with viral cores, as cryo-samples have much worse autofluorescence issue than the room-temperature samples; (3) There could be labeled-IN aggregates not packed into cores; (4) The acquisition failed or the tomogram was of poor quality (technical loss).

## Alignment of tilt series

The frames of each tilt series were corrected for beam-induced motion using MotionCor2 (Zheng et al, 2017). For the alignment and generation of tomograms, tilt series were aligned in IMOD version 4.11 (Kremer et al, 1996) by patch tracking, using patches of 200 × 200 pixels and a fractional overlap of 0.45 in both X and Y. The alignment results were inspected manually, and bad frames were removed. For the experimental group, a total of 269 tomograms were reconstructed at a binning of 6, resulting in 97 tomograms at a pixel size of 11.418 Å/pixel, 167 tomograms at a pixel size of 11.64 Å/pixel, and 5 tomograms at a pixel size of 12 Å/pixel. For the control group of full-genome virions, 58 tomograms were reconstructed at a pixel size of 8.04 Å/pixel. For better visualization, SIRT-like filtering was applied to the reconstructed tomograms with 8 iterations.

## Measurements of HIV-1 cores and NPCs

To measure the diameters of NPCs and the width of HIV-1 cores at different stages of nuclear import, tomograms were reconstructed using IMOD version 4.11.1 with a binning factor of 4. This resulted in pixel sizes of 7.612 Å/pixel, 7.76 Å/pixel, and 8 Å/pixel for the experimental group, and 5.36 Å/pixel for tomograms of the full-genome virion. The central slices of each NPC and HIV-1 core were then imported into ImageJ (Schneider et al, 2012) for precise measurements. Lines were drawn between the measuring points (ONM-INM fusion points for NPCs and the widest part for cores), and gray values were calculated along the lines. The average distance between two dropping points was recorded.

For the experimental group, 467 out of 510 cores were measured, with the remainder being unmeasurable because they partially resided within the tomogram. For full-genome virions, 214 cores were measured. For cores from the initial purification of VLPs, 131 cores were measured directly on the motion-corrected micrographs. A total of 240 NPCs with clearly definable double nuclear envelope boundaries were measured.

## Statistical analyses

To assess the significance of size differences across HIV-1 cores in multiple states, Brown–Forsythe and Welch ANOVA tests were employed. Chi-square test was applied to determine whether the distributions of cone-shaped and tube-shaped HIV-1 cores were independent of their states (Table EV1). Fisher's exact test was used to examine the deformation of NPCs during HIV-1 passage. The significance of size differences in NPCs was assessed using an unpaired two-tailed $t$ test, including 70 HIV-1 occupied NPCs and 159 empty NPCs. The significance of size differences in HIV-1 cores between the experimental and control groups was also assessed using an unpaired $t$ test. All statistical analyses were calculated and plotted using Prism 10.

## Template matching

To localize individual CA hexamers in the tomogram, template matching was performed using emClarity version 1.5.0.2 (Himes and Zhang, 2018; Ni et al, 2022) with non-CTF-corrected tomograms binned at 6×. The procedure utilized a template

derived from EMD-12452 (Ni et al, 2021), which was low-pass filtered to 30 Å. Hexamer peaks were selected with the MagpiEM tool (available at https://github.com/fnight128/MagpiEM), and particle selection was identified based on the geometric constraints of the CA capsid lattice. Any hexamers that did not conform to the expected hexagonal lattice geometry were automatically excluded, followed by manual verification to ensure selection precision. To localize NPC subunits, a 1/8 truncated EM map was generated use the EMD-14321 (Mosalaganti et al, 2022) as the initial template and the template matched coordinates were further cleaned up using MagpiEM. To localize 80S ribosomes and nucleosomes, the same procedure was used without the inspection by MagpiEM, employing low-pass-filtered maps derived from EMD-1780 (Armache et al, 2010) and EMD-16978 (Hou et al, 2023), respectively.

## Subtomogram averaging

The 3D alignment and averaging of the hexameric CA were refined through progressive binning steps from 6 to 2, employing emClarity version 1.5.3.10 (Ni et al, 2022) and maintaining C6 symmetry throughout the alignment process. The final density map at 2x binning was enhanced by sharpening with a B-factor of -10. For the CA hexamer of outside (approaching and docking) HIV-1 VLP cores, eleven tomograms and 1,222 particles were used. For the CA hexamer of traversing HIV-1 VLP cores, eleven tomograms were selected, and 1,085 particles were used. For the CA hexamer of imported HIV-1 VLP cores, nine tomograms were selected, and 1,106 particles were used. The resolution of the reconstructed structure was calculated using a gold-standard Fourier shell correlation (FSC) cut-off of 0.143. The final resolution was determined at 12 Å for outside CA hexamer and 13.5 Å for both traversing and imported CA hexamers (Fig. EV2D). Structural fitting was conducted in ChimeraX (Goddard et al, 2018), enabling detailed comparison of the density maps with the PDB 6SKK (Ni et al, 2020). For the alignment of NPC ring moieties, the box was shifted along the Z axis of the coordinates from template matching for particle extraction, a total of 332 particles were extracted for each moiety. And progressive refinement was performed from bin 12 to 6, the final resolution was determined at 32 Å for cytoplasmic ring (CR) and 38 Å for inner ring (IR) (Fig. EV2D). Due to the low particle number, nuclear ring and luminal ring could not be resolved in this study.

## Segmentation

To enhance the segmentation, reconstructed bin6 tomograms were corrected for the missing wedge and denoised using IsoNet version 0.2 (Liu et al, 2022), applying 35 iterations with sequential noise cut-off levels of 0.05, 0.1, 0.15, 0.2, and 0.25 at iterations 10, 15, 20, 25, and 30, respectively. Membranes and nuclear envelopes in all tomograms were initially segmented using MemBrain-seg (Isensee et al, 2021), then imported into ChimeraX (Goddard et al, 2018) for manual cleaning and polishing. For the top view of nuclear envelopes, segmentation was carried out in Amira (Thermo Fisher Scientific).

Nucleosomes and 80S ribosomes were mapped back to the tomograms with segmented membranes using ChimeraX (Goddard et al, 2018) and ArtiaX (Ermel et al, 2022), based on their positions and orientations after template matching. NPCs were manually placed back based on the positions of holes (occupied by NPCs) on segmented nuclear envelopes. HIV-1 cores were mapped back based on the refined coordinates and orientations of CA hexamers from subtomogram averaging, with missing CA hexamers manually placed back using the low-pass-filtered model EMD-12452 (Ni et al, 2021). For better visualization, 80S ribosomes depicted in the segmented volume were generated by applying a low-pass filter with a cut-off of 15 Å to the original model EMD-1780 (Armache et al, 2010). Nucleosomes were depicted using EMD-16978 (Hou et al, 2023) and NPCs were illustrated using EMD-14321 (Mosalaganti et al, 2022) where the eight subunits could not be fully matched.

## Data availability

All data needed to evaluate the conclusions in the paper are present in the paper and/or the Supplementary information and source data are provided with this paper. The subtomogram averaged maps for the CA hexamers from HIV-1 VLP cores in outside, traversing, and imported states and NPC CR and IR are deposited in the public database EMDB under the accession codes: EMD-53458 (outside), EMD-53459 (traversing), EMD-53460 (imported), EMD-54198 (CR), EMD-54199 (IR). For Figs. 2C,D, 3A,B, 4C, 5A–D, the specific URL and access for the source data deposited online in the public database EMPIAR is under the accession code EMPIAR-12856. The scripts used in this study, and relevant codes are deposited in GitHub [https://github.com/fnight128/MagpiEM] and Zenodo [https://doi.org/10.5281/zenodo.8362772].

The source data of this paper are collected in the following database record: biostudies:S-SCDT-10_1038-S44319-025-00567-6.

## Peer review information

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

## Acknowledgements

We thank Dr. Loic Carrique, Dr. Helen Duyvesteyn, Dr. David Owen, Dr. James Bancroft, Mr. Edward Drydale, and Dr. James Gilchrist for their support in data collection. We thank Dr. Yuta Hikichi and Dr. Eric O Freed for kindly providing the HIV-1 virions. We thank Dr. Juan Shen for her help in the sample shipment and management. We thank Dr. Luiza Mendonça, Dr. Nathan Hardenbrook, and Dr. Joshua Hope for their suggestions in the sample preparation. Dr. Yao Shen is further supported by a CIHR fellowship from the Canadian Institutes of Health Research (funding reference number: 194032), followed by an EMBO fellowship (ALTF 96-2024). We acknowledge The Oxford Particle Imaging Centre (OPIC) for access of cryo-FIB/SEM instruments (Arctis and Aquilos 2) and cryo-EM instrument (Krios). We acknowledge Oxford Cellular Imaging Core Facility (CICF) for access of fluorescence microscopes and imaging analysis software. We acknowledge Diamond Light Source for access and support of the cryo-EM facilities at the UK national electron Bio-Imaging Centre (eBIC), (proposal NT29812). Computation was performed at the Diamond Light Source and Oxford Biomedical Research Computing (BMRC) facility supported by the Wellcome Trust Core Award Grant Number 203141/Z/16/Z with additional support from the NIHR Oxford BRC. This work was supported by US National Institutes of Health grants U54AI170791, R21AI184080, the UK Wellcome Trust Investigator Award 206422/Z/17/Z and the UK Wellcome Discovery Award 311427/Z/24/Z, the UK Biotechnology and Biological Sciences Research Council grant BB/S003339/1, and ERC AdG grant 101021133.

## Author contributions

**Zhen Hou**: Data curation; Formal analysis; Validation; Investigation; Visualization; Methodology; Writing—original draft; Writing—review and editing. **Stanley Fronik**: Formal analysis; Investigation; Methodology; Writing—original draft; Writing—review and editing. **Yao Shen**: Data curation; Formal analysis; Investigation; Visualization; Methodology; Writing—original draft; Writing—review and editing. **Long Chen**: Formal analysis. **Christopher Thompson**: Data curation; Methodology. **Sarah Neumann**: Data curation; Methodology. **Peijun Zhang**: Conceptualization; Resources; Formal analysis; Supervision; Funding acquisition; Visualization; Methodology; Project administration; Writing—review and editing.

Source data underlying figure panels in this paper may have individual authorship assigned. Where available, figure panel/source data authorship is listed in the following database record: biostudies:S-SCDT-10_1038-S44319-025-00567-6.

## Disclosure and competing interests statement

The authors declare no competing interests.

# Expanded View Figures

**Figure EV1. Characterization of HIV-1 cores, CEM cell permebilization, and core-nuclei association using confocal microscopy and cryo-EM.** ▶

(A) Representative confocal images of mechanically permeabilized CEM cells, shown in transillumination channel (left), SiR-DNA channel (middle) and merged with SiR-DNA in magenta (right). (B) Representative confocal images of nuclei of mechanically permeabilized CEM cells, mixed with 150 kDa FITC-dextran, shown in FITC-dextran channel (left), SiR-DNA channel (middle) and merged with SiR-DNA in magenta and FITC in green (right). (C) Two representative cryo-EM images of isolated mature cone-shaped and tube-shaped HIV-1 VLP cores. Inset, mNeonGreen-IN labelled HIV-1 VLP core bands after "spin thru" detergent treatment. The red arrowhead indicates the band extracted for this study. (D) A bar chart showing the distribution of cone-shaped and tube-shaped HIV-1 cores observed within virions and in cores purified from VLPs, respectively. Data were collected from a representative cryo-EM grid. (E) A violin plot showing the width of HIV-1 cores measured from within virions and from cores purified from VLPs, respectively. The width of cone-shape cores (at the wide end) from virions measures $56.79 \pm 4.935$ nm (SE $= 0.3678$, $n = 180$) and that from VLPs measures $56.43 \pm 6.199$ nm (SE $= 0.7573$, $n = 67$). The width of tube-shaped cores from virions measures $41.91 \pm 5.056$ nm (SE $= 0.8672$, $n = 34$) and that from VLPs measures $40.95 \pm 3.314$ nm (SE $= 0.4143$, $n = 64$). Black lines represent the medians, white lines represent the quartiles, and white dots represent individual HIV-1 cores (two-sided $t$ test, ns $=$ no significance). Data were collected from a representative cryo-EM grid. (F) Representative confocal images of nuclei from mechanical lysed (left) or detergent-permeabilized (right) T cells, mixed with isolated HIV-1 VLP cores, shown in single slice (top) and MIP (bottom). HIV-1 cores are labeled with mNeonGreen-IN (green) and nuclei labeled with SiR-DNA (magenta). (G) A box plot representing the number of mNeonGreen-IN puncta decorating a single nucleus obtained either by mechanical lysis or by digitonin permeabilization of CEM cells. The mechanical lysis condition yielded $150 \pm 48$ puncta per nucleus (Fig. 1D) based on two biological replicates: replicate 1: $160 \pm 52$ ($n = 9$) and replicate 2: $136 \pm 42$ ($n = 7$). The digitonin permeabilization condition yielded $217 \pm 77$ puncta per nucleus (Fig. 1D) based on two biological replicates: replicate 1: $243 \pm 93$ ($n = 9$) and replicate 2: $185 \pm 32$ ($n = 7$). In the box plot, the center line is the median, whiskers are min and max, and box bounds are the 25th and 75th percentiles.

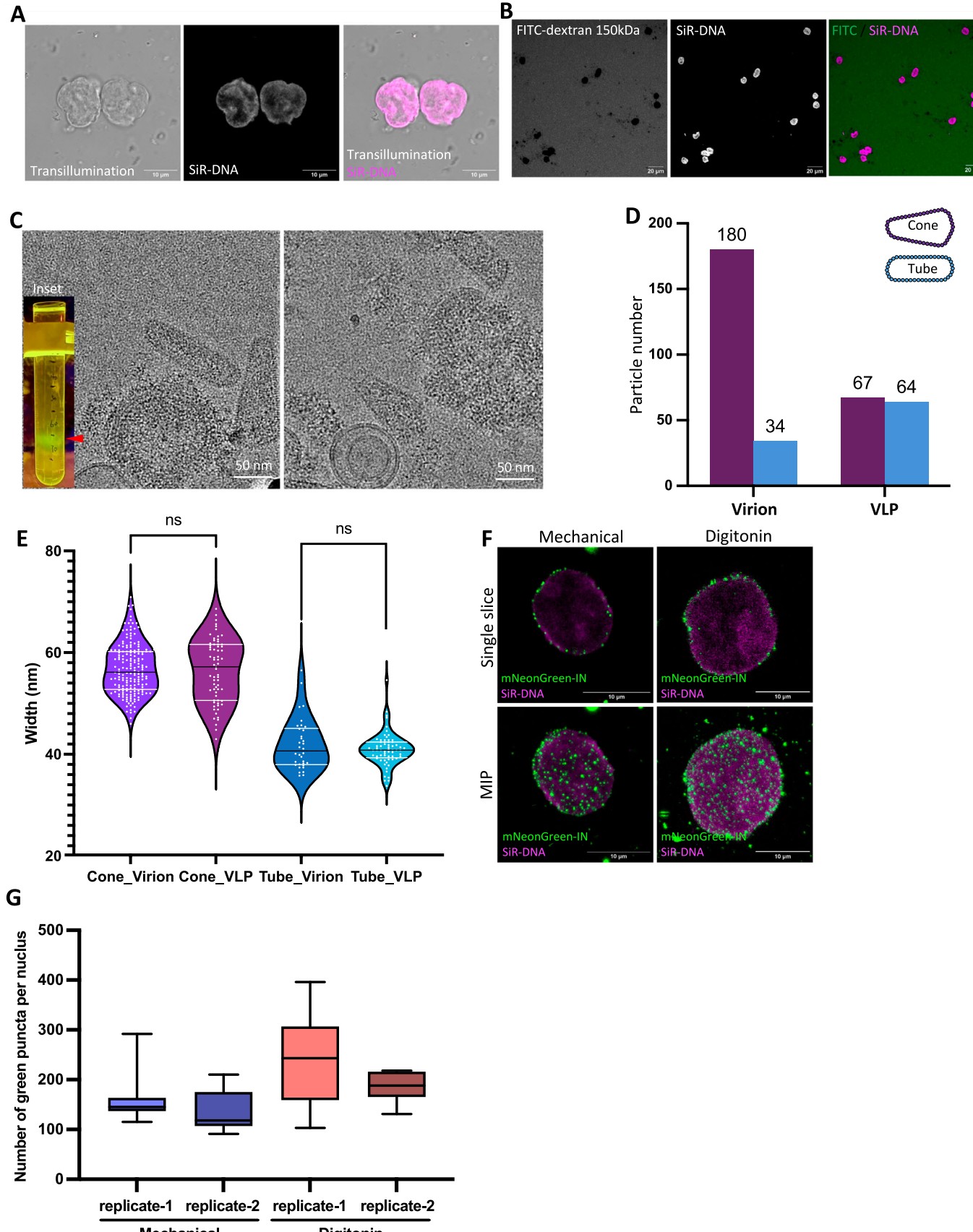

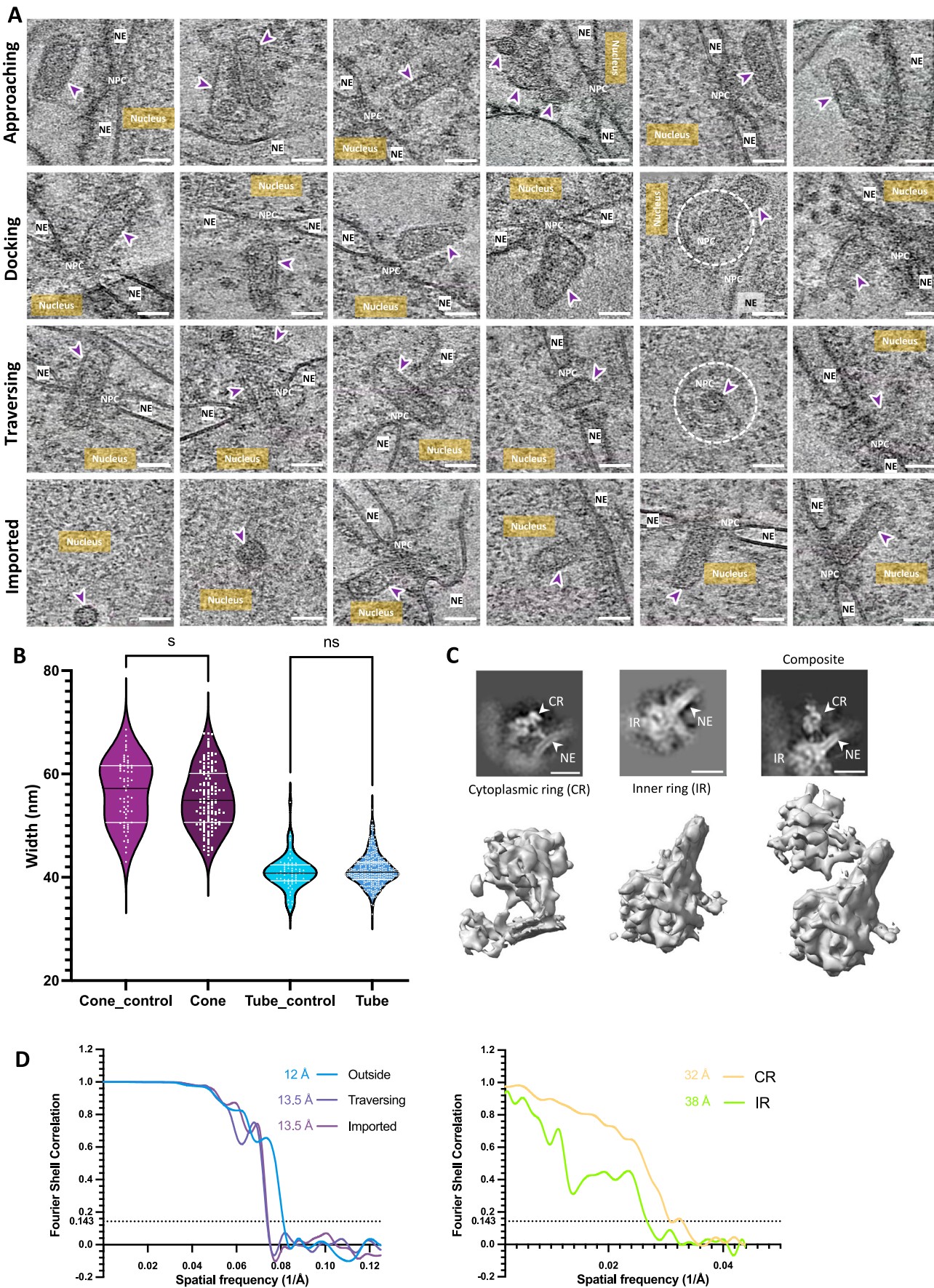

**Figure EV2. Structural characterization of HIV-1 cores during nuclear import and NPC ring moieties by cryo-ET.**

(A) Gallery of HIV-1 cores in multiple states during the nuclear import. Six representative tomographic slices from each state are showcased. HIV-1 cores are indicated by purple arrowheads, the nucleus, NE, and NPC are annotated accordingly. Scale bars = 50 nm. (B) A violin plot of the width of HIV-1 cores associated with nuclei and from the input control, for both cone-shaped and tube-shaped cores. The wide end of cone-shaped HIV-1 cores measures 56.43 ± 6.199 nm (SE = 0.7573, $n = 67$) for the input control, and 55.28 ± 5.730 nm (SE = 0.4968, $n = 133$) for the associated with nuclei. The width of tube-shaped HIV-1 cores measures 40.95 ± 3.314 nm (SE = 0.4143, $n = 64$) for the input control, and 41.43 ± 2.994 nm (SE = 0.1638, $n = 334$) for the associated with nuclei. Black lines represent the medians, white lines represent the quartiles, and white dots represent individual HIV-1 cores (two-sided $t$ test, ns = no significance). The images of input control were collected from a representative grid and the HIV-1 cores associated with nuclei were imaged from 12 grids used for cryo-CLEM. (C) Structures of NPC ring moieties determined by subtomogram averaging. Maps are aligned and contoured to the same level ($3\sigma$). Upper: central slices of the EM maps in the XZ plane; Lower: EM density maps. Cytoplasmic ring, CR; Inner ring, IR. Scale bar, 20 nm. (D) Gold-standard Fourier shell correlation (FSC) curves of subtomogram averaged maps. The resolution is indicated at 0.143 FSC cut-off.

