## [Peer Review File · EMBO Reports]

Direct Visualization of HIV-1 Core Nuclear Import and Its Interplay with the Nuclear Pore

Zhen Hou, Stanley Fronik, Yao Shen, Long Chen, Christopher Thompson, Sarah Neumann, and Peijun Zhang

Corresponding author(s): Peijun Zhang (peijun.zhang@strubi.ox.ac.uk)

Review Timeline:

Submission Date:	4th May 25
Editorial Decision:	23rd Jun 25
Revision Received:	2nd Jul 25
Editorial Decision:	4th Aug 25
Revision Received:	10th Aug 25
Accepted:	15th Aug 25

Editor: Esther Schnapp

Transaction Report:

Dear Peijun,

Thank you for the submission of your manuscript to EMBO reports. We have now received the full set of referee reports that is pasted below.

As you will see, all referees acknowledge that the findings are interesting. Referees 1 and 2 are overall very positive and only have a few comments. Referee 3 is asking for more revisions and I think all points are good and should be addressed. Please let me know in case you disagree and we can discuss the exact revision requirements further, also in a video chat, if you like.

I would thus like to invite you to revise your manuscript with the understanding that the referee concerns must be fully addressed and their suggestions taken on board. Please address all referee concerns in a complete point-by-point response. Acceptance of the manuscript will depend on a positive outcome of a second round of review. It is EMBO reports policy to allow a single round of major revision only and acceptance or rejection of the manuscript will therefore depend on the completeness of your responses included in the next, final version of the manuscript.

We realize that it is difficult to revise to a specific deadline. In the interest of protecting the conceptual advance provided by the work, we recommend a revision within 3 months (23rd Sep 2025). Please discuss the revision progress ahead of this time with the editor if you require more time to complete the revisions.

- 1) A data availability section providing access to data deposited in public databases is missing. If you have not deposited any data, please add a sentence to the data availability section that explains that.
- 2) Your manuscript contains statistics and error bars based on $n=2$. Please use scatter blots in these cases. No statistics should be calculated if $n=2$.

5) a complete author checklist, which you can download from our author guidelines <<https://www.embopress.org/page/journal/14693178/authorguide>>. Please insert information in the checklist that is also reflected in the manuscript. The completed author checklist will also be part of the RPF.

6) Please note that all corresponding authors are required to supply an ORCID ID for their name upon submission of a revised manuscript (<<https://orcid.org/>>). Please find instructions on how to link your ORCID ID to your account in our manuscript tracking system in our Author guidelines <<https://www.embopress.org/page/journal/14693178/authorguide#authorshipguidelines>>

7) Before submitting your revision, primary datasets produced in this study need to be deposited in an appropriate public database (see <https://www.embopress.org/page/journal/14693178/authorguide#datadeposition>). Please remember to provide a reviewer password if the datasets are not yet public. The accession numbers and database should be listed in a formal "Data Availability" section placed after Materials & Method (see also <https://www.embopress.org/page/journal/14693178/authorguide#datadeposition>). Please note that the Data Availability Section is restricted to new primary data that are part of this study. * Note - All links should resolve to a page where the data can be accessed. *
If your study has not produced novel datasets, please mention this fact in the Data Availability Section.

12) All Materials and Methods need to be described in the main text using our 'Structured Methods' format, which is required for all research articles. According to this format, the Methods section includes a separate Reagents and Tools Table file (listing key reagents, experimental models, software and relevant equipment and including their sources and relevant identifiers) and a Methods and Protocols section describing the methods using a step-by-step protocol format. The aim is to facilitate adoption of the methodologies across labs. More information on how to adhere to this format as well as a downloadable template (.docx) for the Reagents and Tools Table can be found in our author guidelines: <https://www.embopress.org/page/journal/14693178/authorguide#structuredmethods>.

An example of a Method paper with Structured Methods can be found here: <https://www.embopress.org/doi/full/10.1038/s44320-024-00037-6#sec-4>

I look forward to seeing a revised form of your manuscript when it is ready.

Referee #1:

In this manuscript, Hou et al. developed a very sophisticated approach to directly visualize HIV-1 capsid and its interactions with nuclear pore complex during nuclear import. Remarkably, the authors were able to capture over 500 rare events which enabled them to uncover extensive remodeling in both viral capsids and nuclear pore complexes. This is a truly amazing work. The manuscript is well written. I am particularly impressed by high-contrast cryo-ET reconstructions from thin lamellae, which were prepared by cryo-FIB milling guided by cryo-CLEM. This study is technically exceptional given so many advanced techniques are well integrated to make this project possible. This study is also biologically significant because the authors were able to generate statistically sufficient data for detailed analyses and reveal remodeling during nuclear import of HIV-1 capsid.

Specific concerns:

1. In "Subtomogram averaging of the HIV-1 capsid lattice during nuclear import" section, three CA hexamer structures were resolved. However, the key message of the paragraph is not clear.
2. Structural remodeling of HIV-1 capsid and NPCs is not well presented in current figures. As an example, the images in Fig.5E are not obvious. In fact, it is difficult to see the NPCs and their differences. Given that subtomogram averaging was used extensively in the manuscript, it is not clear why the resulting averaged structures of NPC were used in segmentations and movies but not in the main Figures to demonstrate the remodeling of NPCs.

Referee #2:

Hou et al developed an integrated in-situ workflow for precise targeting of nuclear import events of HIV1 through the NPC, using cryo-CLEM, cryo-milling and cryo-ET at near-native conditions.

Besides the technical achievement, the work stands out by the large number of observed nuclear capsids (510) and NPC in multiple states (unassociated, approaching, docking, traversing, imported). Robust statistics revealing core morphologies and translocation-associated remodeling. Excellent high-throughput correlative cryo-ET. Morphological characterization showed that tube-shaped cores are preferentially imported, suggesting NPCs act as selective filters. The methods used are broadly applicable to dynamic processes such as viral import of other viruses. Experimental conditions are described in sufficient detail in the methods section.

This work provides unprecedented insights into the structural dynamics of nuclear import of HIV1. The statistical analysis is convincing. The discovery that tube-shaped cores are preferentially imported and the observation of NPC dilation during core translocation are novel.

I have no major concerns. This is excellent, innovative and impactful work, as expected from this leading group.

Fig4E: Please add scale bar

Referee #3:

The manuscript by Hou and colleagues describes a cryo-electron tomography study of purified HIV-1 cores added to isolated cell nuclei. The authors analyse the data to further describe the HIV-1 nuclear import process.

This manuscript can be seen as a follow up on two studies from the Beck/Krausslich labs (Zila et al. 2021 and Kreysing et al 2025). The novelty over the two published studies is that a much larger number of cores could be observed by adding purified cores to nuclei. This approach will be valuable to provide a more complete, statistical understanding of the process.

The authors have completed a technically challenging study, their data is useful (including the parts that are confirmatory), will be of interest to the field, and in my opinion the content is appropriate for publication in EMBO Reports. There are some weaknesses that need to be addressed.

Major comments:

1. What are all the signals which could not be correlated? Is this just technical loss, or are these real integrase signals that are no longer associated with a conical core, for example because they have uncoated?
2. The core preparation seems to diverge from native morphology and further characterization is therefore needed:
 - 1a. It is difficult to follow what has been done. From Figure EV1 and the methods I think that to control the starting material, a single preparation/grid of core purified from virions was analysed, as well as a single preparation/grid of core purified from VLPs? Is that correct? Please add a description of these control experiments. Please make clear whether all statistics are from a single incubation of cores with nuclei.
 - 1b. Please make clear in the main text what is meant by VLP (this is a lentiviral packaging vector, no genome, tagged integrase...). Does this VLP construct have any implications for the properties of the core?
 - 1c. Figure EV1 suggest a tube:cone ratio of 1:5 in cores purified from virions. How does that compare to previous studies in the literature?
 - 1d. Figure EV1 suggests a 1:1 ratio of tube:cone in the VLP preparation, but Table EV1 suggests a 3:1 tube:cone in the total set of cores observed in the dataset? I expected that at least for unassociated cores these ratios should be the same as in the purified core preparation. Why is there a difference here? In general why is the fraction of tubes so high?
 - 1e. Are the cones that are seen in the preparation "normal" or is there more or less variation in the morphology of the cones themselves? This is important to address the question whether the VLP cores are in general more heterogeneous or whether their only unusual feature is the high fraction of tubes.
 - 1f. Do all of the purified cores contain a dense RNP? There is no genome but something should be packaged? Please provide an analysis of this, and state whether it differs between tubes and cones?
3. The conclusion in section 3 of the results "suggesting that core width is an important factor for successful translocation through the NPC" is not supported by the data. I understand that the reduction in width observed through the import process is entirely due to the increased fraction of tubular cores. The results therefore suggest that core morphology is a factor but not core width. To make this statement about core width you would need to remove the confounding effect of morphology from the analysis, for example by measuring only cones.
4. Is it possible that tubular cores appear enriched in the traversing and imported states not because they get in more easily, but because they are present in these states for a longer time than cones. For example, might conical cores traverse more quickly than tubular cores once they get started? Might conical cores undergo more rapid uncoating or other change post import? Equivalently, might tubular cores have a "defect", for example lacking an RNP, that causes them to arrest post-import and therefore become enriched?
5. The section on structural remodelling of HIV-1 capsid during import is poorly supported and should be removed or greatly improved. Weaknesses are: 5a. Only two cores are analysed. 5b. no control is presented to confirm that there are not damaged cores present in the starting material that might get arrested in the pore. 5c. the images of the cores suggest that the template matching is very error prone, even the undamaged parts of the cores do not show a hexagon pattern. 5d. no details or statistics are presented that allow the quality of the template matching to be assessed. This data is overall not strong enough to conclude that the core is damaged during import.
6. The section on structural remodelling of the nuclear pore does not contain enough information. Are the conclusions related to symmetry for empty and unoccupied cores based only on the illustrative examples shown or has a quantitative analysis been performed?
7. The nuclear pore throughout the figures is not a segmentation, but is a structure that has been placed at the pore. Please explain this and provide some images where the pore segmentation and or raw data can be assessed.
8. Please provide much more substantial comparison and discussion of the previous work, in particular the papers by Zila and Kreysing. Please consider the observations related to the cores, but also to the pores. A couple of additional sentences in the introduction would be useful but most importantly, a section needs to be added in the discussion that directly addresses the

similarities and differences of the studies and the results.

9. Please provide a careful consideration of other possible interpretations of the data and a short discussion of the caveats of the study.

Minor comments:

The phrase in the abstract "we visualized 510 HIV-1 VLP cores undergoing the full progression of nuclear import" should be edited since each core is not observed undergoing the full progression (indeed many might be "stuck").

HIV-1 does not have a "unique ability" to infect non-dividing cells.

The Zhao 2013 paper is not the correct reference in the second paragraph of the introduction, this should be a review or better the original literature (consider for example Welker et al., 2000, Ganser et al., 1999 or earlier). Same goes for the reference to Zhao 2013 in the third paragraph.

Please rethink the color scheme in the figures. For example "purple" and "dark purple" don't help to distinguish the two bars. Don't using purple and blue for everything: (tube vs cone, outside vs imported etc)

Some text in figures is too small.

Please add line numbers or at least page numbers to facilitate review.

Point-by-point responses to reviewers' comments

Referee #1 (Remarks to the Author):

In this manuscript, Hou et al. developed a very sophisticated approach to directly visualize HIV-1 capsid and its interactions with nuclear pore complex during nuclear import. Remarkably, the authors were able to capture over 500 rare events which enabled them to uncover extensive remodeling in both viral capsids and nuclear pore complexes. This is a truly amazing work. The manuscript is well written. I am particularly impressed by high-contrast cryo-ET reconstructions from thin lamellae, which were prepared by cryo-FIB milling guided by cryo-CLEM. This study is technically exceptional given so many advanced techniques are well integrated to make this project possible. This study is also biologically significant because the authors were able to generate statistically sufficient data for detailed analyses and reveal remodeling during nuclear import of HIV-1 capsid.

We appreciate the reviewer's positive remarks.

Specific concerns:

1. In "Subtomogram averaging of the HIV-1 capsid lattice during nuclear import" section, three CA hexamer structures were resolved. However, the key message of the paragraph is not clear.

We thank the reviewer for this helpful comment. We agree that the key message of the cryoET subtomogram averaging (STA) section could be made clearer. To improve clarity and narrative flow, we have now combined this section with the subsequent one, "Structural remodeling of HIV-1 capsid and NPCs during import" (Page 6). The main purpose of the STA analysis is to refine the coordinates of individual CA hexamers within the capsid in order to map them back to the original tomograms and assess the structural integrity of the HIV-1 capsid lattice during nuclear import. This integration helps to better convey the rationale and significance of our approach.

2. Structural remodeling of HIV-1 capsid and NPCs is not well presented in current figures. As an example, the images in Fig.5E are not obvious. In fact, it is difficult to see the NPCs and their differences. Given that subtomogram averaging was used extensively in the manuscript, it is not clear why the resulting averaged structures of NPC were used in segmentations and movies but not in the main Figures to demonstrate the remodeling of NPCs.

We thank the reviewer for this suggestion and have included the STA of NPC (revised Fig. EV2C) to analyze the included angles for the adjacent NPC subunits to illustrate the remodeling of NPC (loss of 8-fold symmetry) (revised Fig.5D, E).

Referee #2 (Remarks to the Author):

Hou et al developed an integrated in-situ workflow for precise targeting of nuclear import events of HIV1 through the NPC, using cryo-CLEM, cryo-milling and cryo-ET at near-native conditions.

Besides the technical achievement, the work stands out by the large number of observed nuclear capsids (510) and NPC in multiple states (unassociated, approaching, docking, traversing, imported). Robust statistics revealing core morphologies and translocation-associated remodeling. Excellent high-throughput correlative cryo-ET. Morphological characterization showed that tube-shaped cores are preferentially imported, suggesting NPCs act as selective filters. The methods used are broadly applicable to dynamic processes such as viral import of other viruses. Experimental conditions are described in sufficient detail in the methods section.

This work provides unprecedented insights into the structural dynamics of nuclear import of HIV1. The statistical analysis is convincing. The discovery that tube-shaped cores are preferentially imported and the observation of NPC dilation during core translocation are novel.

I have no major concerns. This is excellent, innovative and impactful work, as expected from this leading group.

We appreciate the reviewer for the positive remarks.

Fig4E: Please add scale bar

We have now included the scale bar (revised Fig. 4E).

Referee #3 (Remarks to the Author):

The manuscript by Hou and colleagues describes a cryo-electron tomography study of purified HIV-1 cores added to isolated cell nuclei. The authors analyse the data to further describe the HIV-1 nuclear import process.

This manuscript can be seen as a follow up on two studies from the Beck/Krausslich labs (Zila et al. 2021 and Kreysing et al 2025). The novelty over the two published studies is that a much larger number of cores could be observed by adding purified cores to nuclei. This approach will be valuable to provide a more complete, statistical understanding of the process.

The authors have completed a technically challenging study, their data is useful (including the parts that are confirmatory), will be of interest to the field, and in my opinion the content is appropriate for publication in EMBO Reports. There are some weaknesses that need to be addressed.

We thank the reviewer for the positive comments.

Major comments:

1. What are all the signals which could not be correlated? Is this just technical loss, or are these real integrase signals that are no longer associated with a conical core, for example because they have uncoated?

For clarification, we only correlate the signals on the periphery and inside of nuclei on the lamellae, which were used to position the target areas in the TEM overview for cryo-ET data collection. There are several reasons for signals which could not be correlated, including 1) The mNeongreen-labeled IN has departed from capsid; 2) The auto-fluorescence signals that are not associated with viral cores, as cryo-samples have much worse autofluorescence issue than the room-temperature samples; 3) There could be labeled-IN aggregates not packed into cores; 4) The acquisition failed or the tomograms was of poor quality (technical loss).

2. The core preparation seems to diverge from native morphology and further characterization is therefore needed:

1a. It is difficult to follow what has been done. From Figure EV1 and the methods I think that to control the starting material, a single preparation/grid of core purified from virions was analysed, as well as a single preparation/grid of core purified from VLPs? Is that correct? Please add a description of these control experiments. Please make clear whether all statistics are from a single incubation of cores with nuclei.

We are grateful for the reviewer's helpful comment. We have clarified the description in both the figure legend (Figure 1 and Figure EV1) and the methods section to make it clear that the results are based on multiple independent experiments and that the statistical analysis are clearly communicated.

1b. Please make clear in the main text what is meant by VLP (this is a lentiviral packaging vector, no genome, tagged integrase...). Does this VLP construct have any implications for the properties of the core?

We have clarified in the main text (Page 3) that the VLPs refer to replication-deficient, virus-like particles generated using a lentiviral packaging vector PsPAX2 (encoding Gag, Pol, Tat, and Rev proteins) and a pVpr-mNeonGreen-IN plasmid for tagged integrase. These VLPs lack the full viral genome and envelope protein but retain the gag, pol, and rev components necessary for forming mature HIV-1 capsids.

While VLPs mimic key structural aspects of the mature HIV-1 core, the absence of the full viral RNA genome and certain accessory proteins may affect core property.

One difference we have noticed is that there is a higher proportion of tubular cores (~50%) in VLPs than in the native cores (~15%) (**Fig. EV1D**), but the average sizes of cores and tubes did not deviate significantly (**Fig. EV1E**). We have also added a brief discussion on this aspect (Page 7).

1c. Figure EV1 suggest a tube:cone ratio of 1:5 in cores purified from virions. How does that compare to previous studies in the literature?

The tube:cone ratio of **15.8%** measured from virions (**Fig. EV1D**) is broadly consistent with earlier reports that quantified the proportion of tube- versus cone-shaped HIV-1 capsid morphologies. Gross et al. (J Virol, 2000) reported 5–10% tube-shaped cores in HIV-1 strain NL4-3 virions. Briggs et al. (EMBO J, 2003) observed 7% tube-shaped cores in HIV-1 strain NL4-3 virions. Similarly, Benjamin et al. (JMB, 2005) found 11.5% tube-shaped cores in virions generated using an R9-based NL4-3 proviral vector. Woodward et al. (J Virol, 2014) reported ~15% of tube-shaped cores using a proviral construct encoding pNLEGFP-BgII. Thus, our observed tube-shaped frequency aligns well with the reported range in the literature.

1d. Figure EV1 suggests a 1:1 ratio of tube:cone in the VLP preparation, but Table EV1 suggests a 3:1 tube:cone in the total set of cores observed in the dataset? I expected that at least for unassociated cores these ratios should be the same as in the purified core preparation. Why is there a difference here? In general why is the fraction of tubes so high?

Table EV1 suggests a 2.3:1 tube:cone, compared to the 1:1 tube:cone in purified VLP core preparation. We could speculate that this discrepancy may stem from the targeted imaging strategy employed in the correlative workflow. Specifically, we selected imaging regions near or at the nuclear envelope. There might be a bias of cores to be more in tubular shape because they are near the nuclear envelope, even though they are not directly associated with NPCs.

The overall high proportion of tube-shaped cores isolated from VLPs is likely related to the use of the psPAX2 packaging system. This system lacks the viral genomic RNA and associated RNP complex, both could influence capsid assembly. In support of this, Woodward et al. (J Virol, 2014) showed that tube-shaped capsids typically lack encapsidated RNP. Furthermore, theoretical modeling studies (Erdemci-Tandogan et al., J. Phys. Chem. B, 2016) have demonstrated that the presence and confinement of the viral genome, along with its interactions with capsid proteins, play a key role in promoting conical capsid formation in mature HIV-1 particles.

Even though our study started with 50% tubular cores, these cores were significantly further enriched through the nuclear import process, making the ratio of tube:cone even higher in the docking and traversing stages, and to near exclusive tubular-shape for the imported cores.

1e. Are the cones that are seen in the preparation "normal" or is there more or less variation in the morphology of the cones themselves? This is important to address

the question whether the VLP cores are in general more heterogeneous or whether their only unusual feature is the high fraction of tubes.

The cone-shaped cores observed in the VLP preparation appear “normal” in terms of size and morphology and are consistent with the cone-shaped core found in virions in this study (**Figure EV1E, no significant difference**) and in previous studies of HIV-1 virions (e.g., Briggs et al., EMBO J 2003; Benjamin et al., JMB 2005). Thus, the primary atypical feature of the VLP cores is a higher proportion of tube-shaped cores. The VLP-derived cores appear not generally more heterogeneous.

1f. Do all of the purified cores contain a dense RNP? There is no genome but something should be packaged? Please provide an analysis of this, and state whether it differs between tubes and cones?

VLPs produced using the psPAX2 packaging system do not package bona fide viral RNPs. However, we did observe internal densities in both cone- and tube-shaped cores, which likely include integrase and nucleocapsid (NC) protein that would bind non-specific viral and cellular RNA. These internal densities are unlikely to represent authentic RNPs.

It is nearly impossible to biochemically separate tubular cores from conical cores, therefore analysis of such differences in packaging of RNA between cones and tubes currently cannot be made. We also argue that a detailed comparison of internal contents between tube- and cone-shaped cores is less relevant to our study, as our focus is on the interaction between the capsid and the nuclear pore complex (NPC). However, we do not exclude the possibility that RNP packaging could influence capsid plasticity, as noted in discussion.

3. The conclusion in section 3 of the results "suggesting that core width is an important factor for successful translocation through the NPC" is not supported by the data. I understand that the reduction in width observed through the import process is entirely due to the increased fraction of tubular cores. The results therefore suggest that core morphology is a factor but not core width. To make this statement about core width you would need to remove the confounding effect of morphology from the analysis, for example by measuring only cones.

We appreciate the reviewer's comments and suggestions.

We respectfully argue that the maximum width of HIV-1 cores, whether tubular or conical, is a critical determinant for nuclear import, as it must be compatible with the inner diameter of the nuclear pore complex (NPC), which closely matches core width. While core morphology may initially appear to influence import (as tubular cores are predominantly observed inside the nucleus), this observation can be more precisely attributed to **differences in maximum width**. Tubular cores, though morphologically distinct, can be viewed as a special case of conical cores with symmetrical ends. Notably, while tubular cores make up approximately 50% of the initial VLP preparation, they are further **enriched within the nucleus**, likely due to their

narrower profile compared to typical conical cores. Supporting this, we also observed two imported **conical cores** whose widths (average 48.6 nm; see initial submission, Page 5) were **much smaller than the population average** (55.28 nm), reinforcing the idea that narrower cores are favored for NPC transit.

4. Is it possible that tubular cores appear enriched in the traversing and imported states not because they get in more easily, but because they are present in these states for a longer time than cones. For example, might conical cores traverse more quickly than tubular cores once they get started? Might conical cores undergo more rapid uncoating or other change post import? Equivalently, might tubular cores have a "defect", for example lacking an RNP, that causes them to arrest post-import and therefore become enriched?

We thank the reviewer for this interesting question. Although it is unlikely that the conical cores traverse more quickly and uncoat more efficiently than the tubular ones as the incubation time in our experiment was 30 minutes and the uncoating of HIV-1 cores was estimated to be hours after the import (Burdick, Ryan C., et al PNAS 2020), we cannot completely rule out this possibility and would like to investigate it in the future. We cannot rule out either if tubular cores have a "defect" that causes them to arrest post-import and therefore become enriched.

5. The section on structural remodelling of HIV-1 capsid during import is poorly supported and should be removed or greatly improved. Weaknesses are: 5a. Only two cores are analysed. 5b. no control is presented to confirm that there are not damaged cores present in the starting material that might get arrested in the pore. 5c. the images of the cores suggest that the template matching is very error prone, even the undamaged parts of the cores do not show a hexagon pattern. 5d. no details or statistics are presented that allow the quality of the template matching to be assessed. This data is overall not strong enough to conclude that the core is damaged during import.

Per reviewer's suggestion, we have now reorganized the sessions, tuning down the structural remodeling of HIV-1 capsid (moved Fig. 5c to Fig. EV2c).

6. The section on structural remodelling of the nuclear pore does not contain enough information. Are the conclusions related to symmetry for empty and unoccupied cores based only on the illustrative examples shown or has a quantitative analysis been performed?

We are grateful for the reviewer's suggestion and have now conducted more thorough quantitative analyses by template matching and subtomogram averaging of NPC subunits. We measured the included angles between adjacent subunits of NPC for the symmetry (or asymmetry) analysis and carried out statistical evaluation. All these are now included in revised **Fig. 5D, E**.

7. The nuclear pore throughout the figures is not a segmentation, but is a structure that has been placed at the pore. Please explain this and provide some images where the pore segmentation and or raw data can be assessed.

We have now included the subtomogram averaging of NPC ring subunits (**Fig. 5D and Fig. EV2C**). However, as not all NPCs could be fully matched with eight subunits, we then used the published structure for the segmentation and have included this information in the figure legends.

8. Please provide much more substantial comparison and discussion of the previous work, in particular the papers by Zila and Kreysing. Please consider the observations related to the cores, but also to the pores. A couple of additional sentences in the introduction would be useful but most importantly, a section needs to be added in the discussion that directly addresses the similarities and differences of the studies and the results.

We have substantially expanded the discussion (Page 6-8) to directly compare our findings with the relevant studies by Zila et al. and Kreysing et al., addressing both similarities and differences in the observations related to HIV-1 cores and nuclear pore complexes (NPCs). Specifically, we now compare the morphology of cores during nuclear import (Page 6-7) and the degree of NPC remodeling observed (Page 7). This comparison highlights how our system complements previous *in situ* studies by enabling higher-throughput structural analysis while preserving key features of the nuclear import process. We have also added a sentence to the introduction (Page 2) to better position our study in the context of prior work.

9. Please provide a careful consideration of other possible interpretations of the data and a short discussion of the caveats of the study.

We have included discussions addressing potential caveats of the study (Page 6-7) as well as alternative interpretations of the data in the main text (Page 5).

Minor comments:

The phrase in the abstract "we visualized 510 HIV-1 VLP cores undergoing the full progression of nuclear import" should be edited since each core is not observed undergoing the full progression (indeed many might be "stuck").

HIV-1 does not have a "unique ability" to infect non-dividing cells.

Revised.

The Zhao 2013 paper is not the correct reference in the second paragraph of the introduction, this should be a review or better the original literature (consider for

example Welker et al., 2000, Ganser et al., 1999 or earlier). Same goes for the reference to Zhao 2013 in the third paragraph.

Revised.

Please rethink the color scheme in the figures. For example "purple" and "dark purple" don't help to distinguish the two bars. Don't using purple and blue for everything: (tube vs cone, outside vs imported etc)

We thank the reviewer for the suggestion. We have tuned the colors to distinguish the two bars but largely keeping the scheme for consistency with depiction of structures.

Some text in figures is too small.

We have now enlarged the small texts.

Please add line numbers or at least page numbers to facilitate review.

Line numbers and page numbers are added.

Dear Prof. Zhang,

Thank you for the submission of your revised manuscript. We have now received the enclosed report from referee 3 who was asked to assess it. The referee points out that some of her/his concerns have not been satisfactorily addressed and lists remaining concerns that must be addressed in a newly revised ms. Please co-submit a point-by-point response with your final ms.

A few editorial requests will also need to be addressed before we can proceed with the official acceptance of your manuscript:

- Your ms has 5 main figures and should therefore be published as a short report with combined results and discussion sections. Please combine the sections, which should also help to reduce the character count to below 20,000 characters (excluding references and methods).

- Please reduce the number of keywords to 5.

- The Code Availability content also needs to be part of the Data Availability Section (DAS).

- The conflict of interest subheading needs to be renamed to "Disclosure and Competing Interests Statement"

- The author credits need to be removed from the ms file. All credits need to be entered during online ms submission.

- This FUNDING INFO is missing in our online submission system: grant R01AI052014 (NIH). Please add.

- The figure legends need to be removed from each Figure file of main and EV figures since the legends are already provided in the ms file, where they should be.

- 4 suppl. tables are provided in one Word file as Table S1-S4; in the ms they are called out as Table EV1-4; since EV tables need to be provided as individual files, each EV table should be uploaded as such and the nomenclature needs to be corrected in all places (source file names, titles in eJP, table legends). "Supplementary information" should not be used; these table callouts need to be corrected.

- 5 movies are uploaded. Their legends should be removed from the ms and each legend should be zipped up with its movie so that 5 zip folders are uploaded. The nomenclature should be corrected to Movie EV1-5 in all places (source file names, titles in eJP, legends, ms callouts).

* Figure Legends - Comments *

- Please note that the exact p values are not provided in the legends of figures 4A, B; 5F. Please provide exact p-values as reasonable.

- Please note that the box plots need to be defined in terms of minima, maxima, centre, bounds of box and whiskers, and percentile in the legend of figure 1D

I would like to suggest a few minor changes to the abstract. Please explain the abbreviations and let me know whether you agree with the changes:

Direct visualization of HIV-1 nuclear import through the nuclear pore complex (NPC) presents a technical challenge due to the rarity of this process. To enable systematic investigation, we developed a robust in situ system that mimics HIV-1 nuclear import in a near-native context using isolated HIV-1 VLP [please spell out VLP] cores and nuclei from CEM [please spell out CEM] cells. This approach supports docking of abundant viral cores at the nuclear envelope and their subsequent translocation into the nucleus. For high-resolution visualization, we implemented an integrated correlative approach to guide cryo-focused ion beam (cryo-FIB) milling and cryo-electron tomography (cryo-ET) imaging, enabling precise targeting and structural characterization of individual nuclear import events. Using this workflow, we visualize 510 HIV-1 VLP cores at distinct stages of nuclear import, capturing key snapshots of the full progression of nuclear import. Subsequent statistical and structural analyses allow classification of core morphologies and identification of translocation-associated remodeling in both viral capsids and nuclear pores. This work provides a methodological foundation for dissecting HIV-1 and potentially other viruses nuclear import processes and post-entry events in a controlled and quantitative manner.

EMBO press papers are accompanied online by A) a short (1-2 sentences) summary of the findings and their significance, B) 2-3 bullet points highlighting key results and C) a synopsis image that is exactly 550 pixels wide and 200-600 pixels high (the height is variable). The synopsis image should provide a sketch of the major findings, like a graphical abstract. Please note that

text needs to be readable at the final size. Please send us this information along with the final manuscript.

Referee #3:

The changes made by the authors have improved the manuscript, and I reiterate my support for this manuscript. Nevertheless, some of my original "major" comments have not been satisfactorily addressed or have been addressed only in the response and not in the manuscript (which helps me but not the reader). I therefore reluctantly ask the authors to work on the manuscript once more before publication. The open points are listed here. I have added an asterisk to the two points that I consider still to be major issues.

Previous comment 1. Please also add clarification of this to the methods section (currently only in response).

Previous comment 2/1a **. I can still not work out easily which experiments have been repeated and whether the data presented represent data from single or multiple experiments. For example: The methods section now states: "HIV-1 core preparations were repeated multiple times, consistently yielding cores of similar quality assessed by cryo-EM imaging". Figure EV1 C and D show plots of core statistics from such preparations, but it is not clear whether these are representative from one preparation, whether they are pooled data from multiple preparations, whether there are differences between preparations. Also for example: Figure 1 legend states "These results are based on analysis from two biological replicates" but not whether the data are pooled between replicates or whether the replicates gave similar results when analysed separately. Please make sure that the reader unfamiliar with the experiments can work out exactly what was done, and whether independent experiments are giving the same results.

Previous comment 2/1d. Please also add clarification to the text that there appears to be an enrichment of tubular cores also prior to nuclear entry if that is the case (currently only in response)

Previous comment 3. The authors argument as presented is not convincing. They consider it likely that width is the critical factor but the text should at least consider the alternative hypothesis that the important factor for entry is morphology (which could itself be associated with RNA content or other factors) and not width. Can this formally be ruled out? A statistical assessment whether the conical cores in the nucleus are narrower than the conical population average would answer this. This only requires a sentence to take this possibility into account.

Previous comment 5 **. This point appears not to have been addressed except by moving the figure to the supplement, which is not an appropriate response to concerns about the data. In my opinion the data is not strong enough to conclude that the core is remodeled during import but the conclusions remain in the manuscript and the abstract. Please do take these comments seriously, it matters to the field whether the core is remodelled during import!

Point-by-point responses to reviewers' comments

Referee #3 (Remarks to the Author):

The changes made by the authors have improved the manuscript, and I reiterate my support for this manuscript. Nevertheless, some of my original "major" comments have not been satisfactorily addressed or have been addressed only in the response and not in the manuscript (which helps me but not the reader). I therefore reluctantly ask the authors to work on the manuscript once more before publication. The open points are listed here. I have added an asterisk to the two points that I consider still to be major issues.

Previous comment 1. Please also add clarification of this to the methods section (currently only in response).

We have now included it in the methods section (page 14).

Previous comment 2/1a **. I can still not work out easily which experiments have been repeated and whether the data presented represent data from single or multiple experiments. For example: The methods section now states: "HIV-1 core preparations were repeated multiple times, consistently yielding cores of similar quality assessed by cryo-EM imaging". Figure EV1 C and D show plots of core statistics from such preparations, but it is not clear whether these are representative from one preparation, whether they are pooled data from multiple preparations, whether there are differences between preparations. Also for example: Figure 1 legend states "These results are based on analysis from two biological replicates" but not whether the data are pooled between replicates or whether the replicates gave similar results when analysed separately. Please make sure that the reader unfamiliar with the experiments can work out exactly what was done, and whether independent experiments are giving the same results.

We thank the reviewer for their suggestions to help improve our data presentation. We have revised the figure legends for Figures EV1C and EV1D, and the methods section (page 13), to state explicitly that the morphology analyses were performed on a single dataset collected from one representative grid, for full-genome virions and VLP cores, respectively. We have revised figure legend for EV2B to clearly state the samples used for the measurements.

To improve the presentation of Figure 1D, we have included 7 additional nuclei from the digitonin-permeabilized CEM cell condition, bringing the total sampling to \$n = 16\$, comparable to the mechanical lysis condition (\$n = 16\$ ). The mechanical lysis condition yielded \$150 \pm 48\$ puncta per nucleus (Figure 1D), with consistent results across biological replicates: replicate 1: \$160 \pm 52\$ (\$n = 9\$ ) and replicate 2: \$136 \pm 42\$ (\$n = 7\$ ) (Figure R1, New Fig. EV1G). The digitonin permeabilization condition yielded \$217 \pm 77\$ puncta per nucleus (Figure 1D), also with similar results between replicates: replicate 1: \$243 \pm 93\$ (\$n = 9\$ ) and replicate 2: \$185 \pm 32\$ (\$n = 7\$ ) (Figure R1). We have revised the figure legends and methods section (page 10) and added Fig. EV1G to reflect these changes.

Figure R1. A box plot representing the number of mNeonGreen-IN puncta decorating a single nucleus, either by mechanical lysis or by digitonin permeabilization of CEM cells. The mechanical lysis condition yielded 150 ± 48 puncta per nucleus (Figure 1D), with similar results across biological replicates: replicate 1: 160 ± 52 ($n = 9$) and replicate 2: 136 ± 42 ($n = 7$). The digitonin permeabilization condition yielded 217 ± 77 puncta per nucleus (Figure 1D), with similar results between replicates: replicate 1: 243 ± 93 ($n = 9$) and replicate 2: 185 ± 32 ($n = 7$) (Figure R1). The centre line is the median, whiskers are min and max, and box bounds are the 25th and 75th percentiles.

Previous comment 2/1d. Please also add clarification to the text that there appears to be an enrichment of tubular cores also prior to nuclear entry if that is the case (currently only in response)

Text was added on page 5 to clarify that tubular cores account for 64% prior to nuclear entry, compared with 50% in the input sample.

Previous comment 3. The authors argument as presented is not convincing. They consider it likely that width is the critical factor but the text should at least consider the alternative hypothesis that the important factor for entry is morphology (which could itself be associated with RNA content or other factors) and not width. Can this formally be ruled out? A statistical assessment whether the conical cores in the nucleus are narrower than the conical population average would answer this. This only requires a sentence to take this possibility into account.

We appreciate the reviewer's comment regarding the potential role of core morphology in nuclear entry. In our dataset, we observed only two conical cores that were imported into the nucleus. These two cores were indeed narrower than the average conical core, however, the sample size was too small to allow for a meaningful statistical assessment. Our data cannot rule out the alternative hypothesis that nuclear entry depends on morphology rather than width, we have revised the text to explicitly acknowledge this possibility and to clarify the limitation in our statistical analysis in the results and discussion section (page 5).

Previous comment 5 **. This point appears not to have been addressed except by moving the figure to the supplement, which is not an appropriate response to concerns about the data. In my opinion the data is not strong enough to conclude that the core is remodeled during import but the conclusions remain in the manuscript and the abstract. Please do take these comments seriously, it matters to the field whether the core is remodelled during import!

We have now removed the EV figure on core remodelling, the related text and videos.

Prof. Peijun Zhang
University of Oxford
Division of Structural Biology
Wellcome Centre for Human Genetics
Roosevelt Drive
Oxford, oxfordshire OX3 7BN
United Kingdom

Dear Prof. Zhang,

I am very pleased to accept your manuscript for publication in the next available issue of EMBO reports. Thank you for your contribution to our journal.

I slightly modified your short summary an bullet points. Please let me know whether you agree with this:

An in situ system that recapitulates HIV-1 nuclear import using isolated HIV-1 cores and permeabilized CD4+ T cells is combined with cryo-electron tomography, enabling high-throughput ultrastructural and statistical analysis of HIV-1 nuclear import events.

- We achieved > 50% cryo-CLEM efficiency, enabling high-throughput ultrastructural and statistical analysis of HIV-1 nuclear import events.
- We identified 510 import events in situ, supporting robust structural and statistical analysis.
- We captured and analyzed five distinct stages of HIV-1 nuclear import.

Also, could you please explain how the synopsis image was generated? Is it entirely based on your data ?

Yours sincerely,
